# DECOMPOSED DIRECT PREFERENCE OPTIMIZATION FOR STRUCTURE-BASED DRUG DESIGN

## ABSTRACT

Diffusion models have achieved promising results for Structure-Based Drug Design (SBDD). Nevertheless, high-quality protein subpocket and ligand data are relatively scarce, which hinders the models' generation capabilities. Recently, Direct Preference Optimization (DPO) has emerged as a pivotal tool for aligning generative models with human preferences. In this paper, we propose DECOMPDPO, a structure-based optimization method aligns diffusion models with pharmaceutical needs using multi-granularity preference pairs. DECOMPDPO introduces decomposition into the optimization objectives and obtains preference pairs at the molecule or decomposed substructure level based on each objective's decomposability. Additionally, DECOMPDPO introduces a physics-informed energy term to ensure reasonable molecular conformations in the optimization results. Notably, DECOMPDPO can be effectively used for two main purposes: (1) fine-tuning pretrained diffusion models for molecule generation across various protein families, and (2) molecular optimization given a specific protein subpocket after generation. Extensive experiments on the CrossDocked2020 benchmark show that DECOMPDPO significantly improves model performance, achieving up to 95.2% Med. High Affinity and a 36.2% success rate for molecule generation, and 100% Med. High Affinity and a 52.1% success rate for molecular optimization.

## 1 INTRODUCTION

Structure-based drug design (SBDD) (Anderson, 2003) is a strategic approach in medicinal chemistry and pharmaceutical research that utilizes 3D structures of biomolecules to guide the design and optimization of new therapeutic agents. The goal of SBDD is to design molecules that bind to specific protein targets. Recent studies viewed this problem as a conditional generative task in a data-driven way, and introduced powerful generative models equipped with geometric deep learning (Powers et al., 2023). For example, Peng et al. (2022); Zhang & Liu (2023) proposed to generate the atoms or fragments sequentially by a SE(3)-equivariant auto-regressive model, while Luo et al. (2021); Peng et al. (2022); Guan et al. (2023a) introduced diffusion models (Ho et al., 2020) to model the distribution of types and positions of ligand atoms. However, the scarcity of high-quality protein-ligand complex data has emerged as a significant bottleneck for the development of generative models in SBDD (Vamathevan et al., 2019). The success of deep learning typically relies on large-scale datasets. In fields like computer vision and natural language processing, the proliferation of the social media has greatly simplified the collection of text, images, and videos, thereby accelerating advancements. In contrast, the collection of protein-ligand binding data is challenging and limited due to the complex and resource-intensive experimental procedures. Notably, the CrossDocked2020 dataset (Francoeur et al., 2020), a widely-used dataset for SBDD, consists of ligands that are docked into multiple similar binding pockets across the Protein Data Bank using docking software. This may be regarded as a form of data augmentation; while it expands the dataset's size, it may unavoidably introduce some low-quality data. As highlighted by Zhou et al. (2024a), the ligands in the CrossDocked2020 dataset have moderate binding affinities, which do not meet the stringent demands of drug design. Moreover, the number of unique ligands remains the same before and after this data augmentation, limiting generative models from learning diverse and high-quality molecules.

To address the aforementioned challenge, Xie et al. (2021); Fu et al. (2022) provided a straightforward method for searching molecules with desired properties in the extensive chemical space. However, pure searching or optimization methods lack generative capabilities and fall short in the diversity of the designed molecules. Zhou et al. (2024a) integrated conditional diffusion models with

iterative optimization by providing molecular substructures as conditions and iteratively replacing the substructures with better ones. This method achieves better properties while maintaining a certain level of diversity. Nonetheless, the performance of this method is still limited due to fixed model parameters during the optimization process.

To break the bottleneck, we propose a method for multi-objective optimization, aligning diffusion models with practical pharmaceutical requirements of drug discovery using generated data. Inspired by the decomposition nature of ligand molecules, DECOMPDPO introduces decomposition into optimization objective to provide more flexibility in preference selection and alignment. Based on each objective's decomposability, DECOMPDPO directly aligns model with preferences using GLOBALDPO with molecule pairs or LOCALDPO with decomposed substructure pairs. Recognizing the importance of maintaining reasonable molecular conformations during optimization, DECOMPDPO integrates physics-informed energy terms to penalize molecules with poor conformations. Additionally, a linear beta schedule is proposed for improving optimization efficiency. We apply DECOMPDPO to two scenarios: structure-based molecule generation and structure-based molecular optimization. Under both settings, DECOMPDPO can significantly outperform baselines, demonstrating its effectiveness. We highlight our contributions as follows:

- We propose DECOMPDPO, which introduces decomposition into the optimization objectives to improve optimization effectiveness and flexibility, directly aligning generative diffusion models with practical pharmaceutical requirements using multi-granularity preferences.
- Our approach is applicable to both structure-based molecular generation and optimization. Notably, DECOMPDPO achieves 95.2% Med. High Affinity and a 36.2% success rate for molecule generation, and 100% Med. High Affinity and a 52.1% success rate for molecular optimization on CrossDocked2020 dataset.
- To the best of our knowledge, we are the first to introduce preference alignment to structure-based drug design. Our approach aligns the generative models for SBDD with the practical requirements of drug discovery.

Recently, an independent concurrent work by Gu et al. (2024) proposed a different framework that also uses preference alignment methods to fine-tune diffusion models for SBDD. Notably, they regularized the DPO objective to alleviate overfitting on the winning data. However, they primarily focused on optimizing affinity-related metrics for molecule generation and did not evaluate optimized molecular conformations, which is an important aspect in drug design. Compared to Gu et al. (2024), we directly formulate preference alignment in SBDD as a multi-objective optimization problem, which is more aligned with pharmaceutical needs, and introduce decomposition into the optimization objectives to provide more flexibility in multi-objective optimization. In addition, we incorporate physics-informed energy terms to penalize unreasonable molecular conformations, thereby maintaining desirable conformations during optimization. Moreover, we demonstrate the effectiveness of DECOMPDPO in molecular optimization through iterative fine-tuning, achieving superior performance compared to existing optimization methods.

## 2 RELATED WORK

**Structure-based Drug Design**  Structure-based drug design (SBDD) aims to design ligand molecules that can bind to specific protein targets. Recent efforts have been made to enhance the efficiency of generating molecules with desired properties. Ragoza et al. (2022) employed variational autoencoder to generate 3D molecules in atomic density grids. Luo et al. (2021); Peng et al. (2022); Liu et al. (2022) adopted an autoregressive approach to generate 3D molecules atom by atom, while Zhang et al. (2022) proposed to generate 3D molecules by predicting a series of molecular fragments in an auto-regressive way. Guan et al. (2023a); Schneuing et al. (2022); Lin et al. (2022) introduced diffusion models to SBDD, which first generate the types and positions of atoms and subsequently determine bond types by post-processing. Some recent studies have sought to further improve efficacy of SBDD methods by incorporating biochemical prior knowledge. DecompDiff (Guan et al., 2023b) proposed decomposing ligands into substructures and generating atoms and bonds simultaneously using diffusion models with decomposed priors and validity guidance. DrugGPS (Zhang & Liu, 2023) considered subpocket-level similarities, augmenting molecule generation through global interaction between subpocket prototypes and molecular motifs. IPDiff (Huang et al., 2023) addressed the inconsistency between forward and reverse processes using a pre-trained protein-ligand interaction prior network. In addition to simply generative modeling of existing protein-ligand pairs, some researchers leveraged optimization algorithms to design molecules

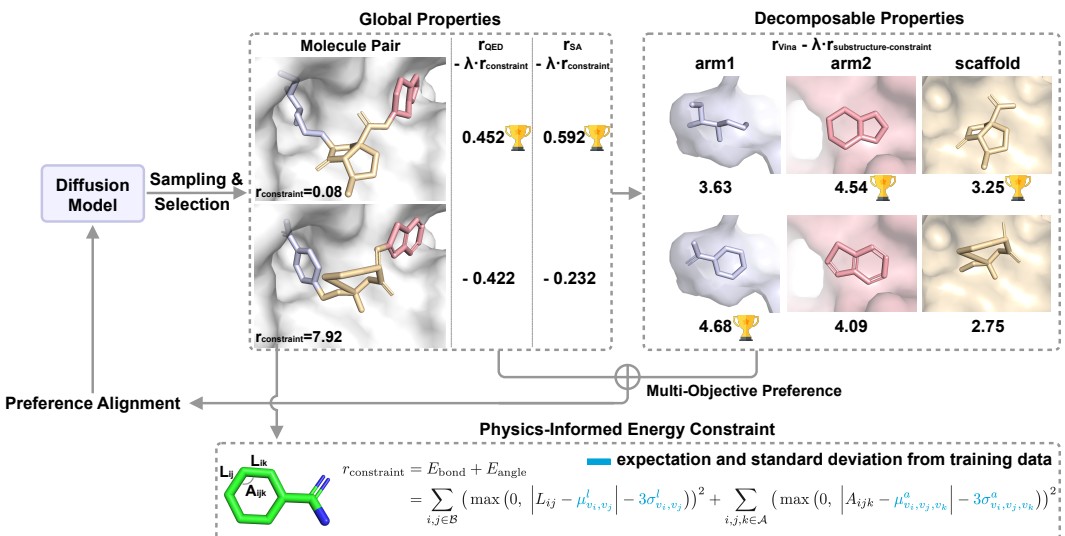

Figure 1: Illustration of DECOMPDPO. This process can be summarized as: (a) Sample molecules and select molecule pairs for each target protein using a pre-trained diffusion model; (b) Construct physically constrained preference for each optimization objective based on its decomposability; (c) Compute the DECOMPDPO loss and align the diffusion model with the multi-objective preference.

with desired properties. AutoGrow 4 (Spiegel & Durrant, 2020) and RGA (Fu et al., 2022) optimized the binding affinity of ligand molecules towards specific targets by elaborate genetic algorithms. RGA (Fu et al., 2022) viewed the evolutionary process as a Markov decision process and guided it by reinforcement learning. EvoSBDD (Reidenbach, 2024) performd an evolutionary algorithm in a pretrained 1D latent space using an AutoDock Vina redocking oracle to optimize generated SMILES. TacoGFN (Shen et al., 2023) uses a Generative Flow Network to optimize 2D molecular graphs for SBDD as a reinforcement learning task. PILOT (Cremer et al., 2024) employs an importance sampling scheme during inference to reweight trajectories during generation and optimize towards targeted objectives. DecompOpt (Zhou et al., 2024a) proposed a controllable and decomposed diffusion model that can generate ligand molecules conditioning on both protein subpockets and reference substructures, and combined it with iterative optimization to improve desired properties by iteratively generating molecules given substructures observed in previous iterations. Our work also focuses on designing molecules with desired properties by optimization. Differently, we optimize the parameters of the model that generate molecules instead of molecules themselves, which has been demonstrated to be more effective and efficient.

**Learning from Human/AI Feedback** Maximizing likelihood optimization of generative models cannot always satisfy users' preferences. Thus introducing human or AI assessment to improve the performance of generative models has attracted significant attention. Reinforcement learning from human/AI feedback (Ziegler et al., 2019; Stiennon et al., 2020; Ouyang et al., 2022; Lee et al., 2023; Bai et al., 2022) was proposed to align large language models to human preference, consisting of reward modeling from comparison data annotated by human or AI and then using policy-gradient methods (Christiano et al., 2017; Schulman et al., 2017) to fine-tune the model to maximize the reward. Similar techniques have also been introduced to diffusion models for text-to-image generation (Black et al., 2023; Fan et al., 2024; Zhang et al., 2024), where the generative process is viewed as a multi-step Markov decision process and policy-gradient methods can be then applied to fine-tuning the models. Recently, Direct Preference Optimization (DPO) (Rafailov et al., 2024), which aligns large language models to human preferences by directly optimizing on human comparison data, has attracted much attention. Wallace et al. (2023) re-formulated DPO and derived Diffusion-DPO for aligning text-to-image diffusion models. The above works focus on aligning large language models or text-to-image diffusion models with human preferences. Recently, Zhou et al. (2024b) proposed to fine-tune diffusion models for antibody design by DPO and choose low Rosetta energy as preference. In our work, we introduce preference alignment to improve the desired properties of generated molecules given specific protein pockets and propose specialized methods to improve the performance of DPO in the scenario of SBDD.

## 3 METHOD

In this section, we present our method, DECOMPDPO, which aligns diffusion models with pharmaceutical needs using physically constrained multi-granularity preferences (Figure 1). We first define the SBDD task and introduce the decomposed diffusion model for this task in Section 3.1. Then, we incorporate decomposition into the optimization objectives and propose DECOMPDPO for multi-objective optimization in Section 3.2. Recognizing the importance of maintaining reasonable molecular conformations during optimization, we introduce physics-informed energy terms for penalizing the reward in Section 3.3. Finally, we introduce a linear beta schedule to improve the efficiency of optimizing diffusion models (Section 3.4).

### 3.1 PRELIMINARIES

In the context of SBDD, generative models are conditioned on the protein binding site, represented as $\mathcal{P} = \{(\boldsymbol{x}_i^{\mathcal{P}}, \boldsymbol{v}_i^{\mathcal{P}})\}_{i \in \{1, \cdots, N_{\mathcal{P}}\}}$, to generate ligands $\mathcal{M} = \{(\boldsymbol{x}_i^{\mathcal{M}}, \boldsymbol{v}_i^{\mathcal{M}}, \boldsymbol{b}_{ij}^{\mathcal{M}})\}_{i,j \in \{1, \cdots, N_{\mathcal{M}}\}}$ that bind to this site. Here, $N_{\mathcal{P}}$ and $N_{\mathcal{M}}$ are the number of atoms in the protein and ligand, respectively. For both proteins and ligands, $\boldsymbol{x} \in \mathbb{R}^3$, $\boldsymbol{v} \in \mathbb{R}^h$, $\boldsymbol{b}_{ij} \in \mathbb{R}^5$ represents the coordinates of atoms, the types of atoms, and the bonds between atoms. Here we consider $h$ types of atoms (i.e., H, C, N, O, S, Se) and 5 types of bonds (i.e., non-bond, single, double, triple, aromatic).

Following the decomposed diffusion model introduced by Guan et al. (2023b), each ligand is decomposed into fragments $\mathcal{K}$, comprising several arms $\mathcal{A}$ connected by at most one scaffold $\mathcal{S}$ ($|\mathcal{A}| \geq 1, |\mathcal{S}| \leq 1, K = |\mathcal{K}| = |\mathcal{A}| + |\mathcal{S}|$). Based on the decomposed substructures, informative data-dependent priors $\mathbb{O}_{\mathcal{P}} = \{\boldsymbol{\mu}_{1:K}, \boldsymbol{\Sigma}_{1:K}, \mathbf{H}\}$ are estimated from atom positions by maximum likelihood estimation, where $\boldsymbol{\mu}_k \in \mathbb{R}^3$ represents the prior center, $\boldsymbol{\Sigma}_k \in \mathbb{R}^{3 \times 3}$ represents the prior covariance matrix, and $\mathbf{H} = \{\eta^{\mathcal{P}} \in \{0,1\}^{N_M \times K} | \sum_{k=1}^K \eta_{ik}^{\mathcal{P}} = 1\}$ represents the prior-atom mapping. This data-dependent prior enhances the training efficacy of the diffusion model, where $\mathcal{M}$ is gradually diffused with a fixed schedule $\{\lambda_t\}_{t=1, \cdots, T}$. We denote $\alpha_t = 1 - \lambda_t$ and $\bar{\alpha}_t = \prod_{s=1}^t \alpha_t$. The $i$-th atom position is shifted to its corresponding prior center: $\tilde{\mathbf{x}}_t^i = \mathbf{x}_t^i - (\mathbf{H}^i)^\top \boldsymbol{\mu}$. The noisy data distribution at time $t$ derived from the distribution at time $t-1$ is computed as follows:

$$p(\tilde{\mathbf{x}}_t | \tilde{\mathbf{x}}_{t-1}, \mathcal{P}) = \prod_{i=1}^{N_{\mathcal{M}}} \mathcal{N}(\tilde{\mathbf{x}}_t^i; \tilde{\mathbf{x}}_{t-1}^i, \lambda_t (\mathbf{H}^i)^\top \boldsymbol{\Sigma}), \tag{1}$$

$$p(\mathbf{v}_t | \mathbf{v}_{t-1}, \mathcal{P}) = \prod_{i=1}^{N_{\mathcal{M}}} \mathcal{C}(\mathbf{v}_t^i | (1 - \lambda_t) \mathbf{v}_{t-1}^i + \lambda_t / K_a), \tag{2}$$

$$p(\mathbf{b}_t | \mathbf{b}_{t-1}, \mathcal{P}) = \prod_{i=1}^{N_{\mathcal{M}} \times N_{\mathcal{M}}} \mathcal{C}(\mathbf{b}_t^i | (1 - \lambda_t) \mathbf{b}_{t-1}^i + \lambda_t / K_b), \tag{3}$$

where $K_a$ and $K_b$ represent the number of atom types and bond types used for featurization. The perturbed structure is then fed into the prediction model, then the reconstruction loss at the time $t$ can be derived from the KL divergence as follows:

$$L^{(x)} = \mathbb{E}_t \left[ ||\mathbf{x}_0 - \hat{\mathbf{x}}_0||^2 \right], \ L^{(v)} = \mathbb{E}_t \left[ \sum_{k=1}^{K_a} \boldsymbol{c}(\mathbf{v}_t, \mathbf{v}_0)_k \log \frac{\boldsymbol{c}(\mathbf{v}_t, \mathbf{v}_0)_k}{\boldsymbol{c}(\mathbf{v}_t, \hat{\mathbf{v}}_0)_k} \right], \ L^{(b)} = \mathbb{E}_t \left[ \sum_{k=1}^{K_b} \boldsymbol{c}(\mathbf{b}_t, \mathbf{b}_0)_k \log \frac{\boldsymbol{c}(\mathbf{b}_t, \mathbf{b}_0)_k}{\boldsymbol{c}(\mathbf{b}_t, \hat{\mathbf{b}}_0)_k}, \right]$$

where $(\mathbf{x}_0, \mathbf{v}_0, \mathbf{b}_0)$, $(\mathbf{x}_t, \mathbf{v}_t, \mathbf{b}_t)$, $(\hat{\mathbf{x}}_0, \hat{\mathbf{v}}_0, \hat{\mathbf{b}}_0)$, represent true atoms positions, types, and bonds types at time 0, time $t$, predicted atoms positions, types, and bonds types at time $t \sim U[0, T]$; $\boldsymbol{c}$ denotes mixed categorical distribution with weight $\bar{\alpha}_t$ and $1 - \bar{\alpha}_t$. The overall loss is $L = L^{(x)} + \gamma_v L^{(v)} + \gamma_b L^{(b)}$, with $\gamma_v, \gamma_b$ as weights of reconstruction loss of atom and bond type. We provide more details for the model architecture in Appendix C. To better illustrate decomposition, we show a decomposed molecule with the arms highlighted in Figure 2.

### 3.2 DIRECT PREFERENCE OPTIMIZATION IN DECOMPOSED SPACE

**Decomposable Optimization Objectives** In realistic pharmaceutical scenarios, potential drug molecules should possess multiple desirable properties, which is rare among all known drug-like molecules. As illustrated in previous work (Zhou et al., 2024a), directly learning the distribution from training data is suboptimal, making it inefficient in generating desired molecules. Despite DecompOpt (Zhou et al., 2024a) fully exploits the power of conditional diffusion models through

iterative generation, the model's upper limit remains constrained by the static parameters learned from offline data. Direct preference optimization offers a simple yet efficient way to align models directly with pairwise preferences. Inspired by the success of introducing decomposition in the drug space (Guan et al., 2023b; Zhou et al., 2024a), we introduce the concept of decomposition into optimization objectives in DECOMPDPO, providing greater flexibility in preference selection and alignment.

We define an optimization objective as decomposable if the property of a molecule is proportional to the sum of the properties of its decomposed substructures. This means that a substructure with a higher property will lead the molecule to have a higher overall property. For example, *Vina Minimize Score* is largely based on pairwise atomic interactions, with each substructure contributing its own set of interactions with the protein target and negligible inter-substructure interactions, making it decomposable. As shown in Figure 2, we validated the proportional relationship of *Vina Minimize Score* in our training dataset. Unfortunately, not every optimization objective is decomposable. Molecular properties such as *QED* and *SA* are non-decomposable, as their calcula-

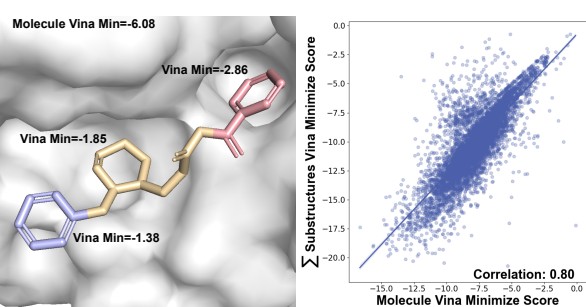

Figure 2: Illustration of decomposable objectives. Decomposition of a molecule into two arms (purple and pink) and a scaffold (yellow), where the sum of the substructures' Vina Minimize Scores equals to the molecule's (left). The Pearson correlation between molecule's and sum of substructure's Vina Minimize Scores in the training dataset (right).

tions involve non-linear operations. We provide more statistical evidence in Appendix D.

**GLOBALDPO** To align the model with practical pharmaceutical preferences, following RLHF (Ouyang et al., 2022), the pre-trained model is fine-tuned by maximizing certain reward functions with the Kullback–Leibler (KL) divergence regularization:

$$\max_{p_\theta} \mathbb{E}_{\mathcal{P}\sim\mathcal{D},\mathcal{M}\sim p_\theta(\mathcal{M}|\mathcal{P})} r(\mathcal{M},\mathcal{P}) - \beta D_{\mathrm{KL}}\left[p_\theta(\mathcal{M} \mid \mathcal{P}) \parallel p_{\mathrm{ref}}(\mathcal{M} \mid \mathcal{P})\right], \quad (4)$$

where $\beta > 0$ is a hyperparameter controlling the deviation from the reference model $p_{\mathrm{ref}}$. Recently, Rafailov et al. (2024) proposed Direct Preference Optimization (DPO), deriving the DPO training loss from the RLHF loss and providing a simpler way to fine-tune the model with pairwise preference data:

$$\mathcal{L}_{\mathrm{DPO}} = -\mathbb{E}_{(\mathcal{P},\mathcal{M}^+,\mathcal{M}^-)\sim\mathcal{D}} \log\sigma\left(\beta\log\frac{p_\theta(\mathcal{M}^+ \mid \mathcal{P})}{p_{\mathrm{ref}}(\mathcal{M}^+ \mid \mathcal{P})} - \beta\log\frac{p_\theta(\mathcal{M}^- \mid \mathcal{P})}{p_{\mathrm{ref}}(\mathcal{M}^- \mid \mathcal{P})}\right). \quad (5)$$

Here, $\mathcal{M}^+$ and $\mathcal{M}^-$ represent the preferred and less preferred molecules, respectively.

As $p_\theta(\mathcal{M}_0 \mid \mathcal{P})$ is intractable for diffusion models, following Diffusion-DPO (Wallace et al., 2023), we define the reward over the entire diffusion process $r(\mathcal{M},\mathcal{P}) = \mathbb{E}_{p_\theta(\mathcal{M}_{1:T}|\mathcal{M}_0,\mathcal{P})}[R(\mathcal{M}_{1:T},\mathcal{P})]$, where $\mathcal{M}_{1:T}$ denotes the diffusion trajectories from the reverse process $p_\theta$. Consequently, the DPO loss is reframed as:

$$\mathcal{L}_{\mathrm{Diffusion\text{-}DPO}} = -\mathbb{E}_{(\mathcal{P},\mathcal{M}^+,\mathcal{M}^-)\sim\mathcal{D}} \log\sigma\left(\beta\mathbb{E}_{\mathcal{M}_{1:T}^+,\mathcal{M}_{1:T}^-}\left[\log\frac{p_\theta(\mathcal{M}_{1:T}^+|\mathcal{P})}{p_{\mathrm{ref}}(\mathcal{M}_{1:T}^+|\mathcal{P})} - \log\frac{p_\theta(\mathcal{M}_{1:T}^-|\mathcal{P})}{p_{\mathrm{ref}}(\mathcal{M}_{1:T}^-|\mathcal{P})}\right]\right). \quad (6)$$

Following Wallace et al. (2023), we further approximate reverse probability $p_\theta$ with forward probability $q$, and utilize Jensen's inequality to externalize the expectation:

$$\mathcal{L}_{\mathrm{Diffusion\text{-}DPO}} = -\mathbb{E}_{\substack{(\mathcal{P},\mathcal{M}^+,\mathcal{M}^-)\sim\mathcal{D},t\sim\mathcal{U}(0,T),\\ \mathcal{M}_t^+\sim q(\mathcal{M}_t^+|\mathcal{M}_0^+),\\ \mathcal{M}_t^-\sim q(\mathcal{M}_t^-|\mathcal{M}_0^-)}} \log\sigma\left(\beta\left[\log\frac{p_\theta(\mathcal{M}_{t-1}^+|\mathcal{M}_t^+,\mathcal{P})}{p_{\mathrm{ref}}(\mathcal{M}_{t-1}^+|\mathcal{M}_t^+,\mathcal{P})} - \log\frac{p_\theta(\mathcal{M}_{t-1}^-|\mathcal{M}_t^-,\mathcal{P})}{p_{\mathrm{ref}}(\mathcal{M}_{t-1}^-|\mathcal{M}_t^-,\mathcal{P})}\right]\right). \quad (7)$$

The Diffusion-DPO loss is applied to align our model with non-decomposable optimization objectives using molecule-level preferences. For clarity, we refer to this molecule-level preference alignment as GLOBALDPO hereafter.

**LOCALDPO** According to the decomposition in drug space, the probability of a molecule is equivalent to the product of the probabilities of its decomposed substructures. As a result, we reformulate the Diffusion-DPO loss as:

$$\mathcal{L}_{\text{DIFFUSION-DPO}} = -\mathbb{E}_{\substack{(\mathcal{P},\mathcal{M}^+,\mathcal{M}^-)\sim\mathcal{D},t\sim\mathcal{U}(0,T),\\ \mathcal{M}_t^+\sim q(\mathcal{M}_t^+|\mathcal{M}_0^+),\mathcal{M}_t^-\sim q(\mathcal{M}_t^-|\mathcal{M}_0^-)}} \log\sigma\left(\beta\sum_i^K\left[\log\frac{p_\theta(\mathcal{M}_{t-1}^{(i)+}|\mathcal{M}_t^{(i)+},\mathcal{P})}{p_{\text{ref}}(\mathcal{M}_{t-1}^{(i)+}|\mathcal{M}_t^{(i)+},\mathcal{P})} - \log\frac{p_\theta(\mathcal{M}_{t-1}^{(i)-}|\mathcal{M}_t^{(i)-},\mathcal{P})}{p_{\text{ref}}(\mathcal{M}_{t-1}^{(i)-}|\mathcal{M}_t^{(i)-},\mathcal{P})}\right]\right), \quad (8)$$

where $\mathcal{M}^{(i)}$ represents the $i$-th decomposed substructure, $\mathcal{M}_t^{(i)+}$ is decomposed from winning molecule, and $\mathcal{M}_t^{(i)-}$ is decomposed from losing molecule: $\mathcal{M}_t^+ = \bigcup_i^K\mathcal{M}_t^{(i)+}, \mathcal{M}_t^- = \bigcup_i^K\mathcal{M}_t^{(i)-}$. Here, the decomposed substructures of the preferred molecule are always considered the winning side, even though it is not always the case that they have better properties.

For decomposable optimization objectives, we incorporate decomposition into preference alignment by directly constructing preference pairs based on the properties of substructures. Using substructure-level preferences, we derive the training loss for LOCALDPO as follows:

$$\mathcal{L}_{\text{LOCALDPO}} = -\mathbb{E}_{\substack{(\mathcal{P},\mathcal{M}^+,\mathcal{M}^-)\sim\mathcal{D},t\sim\mathcal{U}(0,T),\\ \mathcal{M}_t^+\sim q(\mathcal{M}_t^+|\mathcal{M}_0^+),\mathcal{M}_t^-\sim q(\mathcal{M}_t^-|\mathcal{M}_0^-),}} \log\sigma\left(\beta\sum_i^K \text{sign}(r(\mathcal{M}^{(i)+})-r(\mathcal{M}^{(i)-}))\big[A^{(i)}\big]\right),$$

$$\text{where } A^{(i)} = \log\frac{p_\theta(\mathcal{M}_{t-1}^{(i)+}\mid\mathcal{M}_t^{(i)+},\mathcal{P})}{p_{\text{ref}}(\mathcal{M}_{t-1}^{(i)+}\mid\mathcal{M}_t^{(i)+},\mathcal{P})} - \log\frac{p_\theta(\mathcal{M}_{t-1}^{(i)-}\mid\mathcal{M}_t^{(i)-},\mathcal{P})}{p_{\text{ref}}(\mathcal{M}_{t-1}^{(i)-}\mid\mathcal{M}_t^{(i)-},\mathcal{P})}, \quad (9)$$

where $r(\mathcal{M}^{(i)})$ represents the reward of the decomposed substructure $\mathcal{M}^{(i)}$, $\mathcal{M}_t^+ = \bigcup_i^K\mathcal{M}_t^{(i)+}, \mathcal{M}_t^- = \bigcup_i^K\mathcal{M}_t^{(i)-}$. Compared to Diffusion-DPO, LOCALDPO behaves differently when the substructure-level preference is inconsistent with the molecule-level preference, that is when the sign function yields $\text{sign}(\cdot) < 0$. In such cases, if the model's preference, denoted as $A^{(i)}$ in Equation (9), conflict with the substructure-level preference, the loss for LOCALDPO increases because the function $-\log\sigma$ is monotonically decreasing. Conversely, if the model's preference aligns with the substructure-level preference, the loss decreases. As a result, LOCALDPO more effectively corrects misaligned substructure-level preferences.

In multi-objective optimization, different objectives can interfere with each other, leading to suboptimal results. By leveraging decomposed preferences, LOCALDPO offers more flexible and diverse optimization pathways, indirectly mitigating conflicts inherent in multi-objective optimization and enhancing overall performance.

**DECOMPDPO** Based on GLOBALDPO and LOCALDPO introduced above, we construct preference pairs for each optimization objective according to its decomposability. By taking a weighted sum of all the preference alignment losses, we derive the overall loss for DECOMPDPO:

$$\mathcal{L}_{\text{DECOMPDPO}} = \sum_{i\in\mathcal{Q}_{\text{Decomp}}} w_i\mathcal{L}_{\text{LOCALDPO}}(i) + \sum_{j\in\mathcal{Q}_{\text{Non-Decomp}}} w_j\mathcal{L}_{\text{GLOBALDPO}}(j), \quad (10)$$

where $\mathcal{Q}_{\text{Decomp}}$ and $\mathcal{Q}_{\text{Non-Decomp}}$ represent the decomposable and non-decomposable properties, and $w_i, w_j$ are weighting coefficients. This dual-granularity alignment allows for more precise control over the optimization process and offers greater flexibility in selecting preferences to meet the diverse requirements of molecular design.

### 3.3 PHYSICALLY CONSTRAINED OPTIMIZATION

An important aspect of preference alignment in drug design is to maintain reasonable molecular conformations that obey physical rules. Inspired by Wu et al. (2022), we define physics-informed energy terms that penalize bonds and angles which deviate significantly from empirical values, formulated as:

$$E_{\text{bond}} = \sum_{i,j\in\mathcal{B}} \big(\max\big(0, \big|L_{ij} - \mu_{v_i,v_j}^l\big| - 3\sigma_{v_i,v_j}^l\big)\big)^2, \quad (11)$$

$$E_{\text{angle}} = \sum_{i,j,k\in\mathcal{A}} \big(\max\big(0, \big|A_{ijk} - \mu_{v_i,v_j,v_k}^a\big| - 3\sigma_{v_i,v_j,v_k}^a\big)\big)^2, \quad (12)$$

where $\mathcal{B}$ denotes the set of bonds in the molecule, and $\mathcal{A}$ denotes the set of angles formed by two neighboring bonds in $\mathcal{B}$. Here, $L_{ij}$ is the bond length between atoms $i$ and $j$, and $A_{ijk}$ is the radian

of the angle formed by atoms $i$, $j$, and $k$; $\mu^l_{v_i, v_j}$ and $\sigma^l_{v_i, v_j}$ are the expectation and standard deviation of bond lengths between atom types $v_i$ and $v_j$, obtained from the training data. Similarly, $\mu^a_{v_i, v_j, v_k}$ and $\sigma^a_{v_i, v_j, v_k}$ represent the expectation and standard deviation of bond angles formed by atom types $v_i$, $v_j$, and $v_k$. The overall energy term is defined as $r_{\text{constraint}} = E_{\text{bond}} + E_{\text{angle}}$. To prevent the model from learning unrealistic molecular conformations, we adjust the reward by penalizing it with this energy term: $r^*(\mathcal{M}, \mathcal{P}) = r(\mathcal{M}, \mathcal{P}) - \lambda r_{\text{constraint}}(\mathcal{M}, \mathcal{P})$, where $\lambda$ is a weighting factor that balances the importance of the physical constraints.

### 3.4 LINEAR BETA SCHEDULE

Drawing from Equation (4), the parameter $\beta$ serves as a form of regularization, balancing the exploration of high-quality molecules with adherence to the pre-learned prior distribution. During the diffusion sampling process, earlier steps influence the subsequent ones. Moreover, the final few steps are crucial in determining the atoms' types and positions, which significantly affect the molecules' properties and make optimization in the later steps more critical. To improve optimization efficiency, we propose a linear beta schedule $\beta_t = \frac{t}{T}\beta_T$, where $\beta_T$ is the beta parameter at the final time step $T$ of the reverse process. This schedule ensures a progressive reduction in the impact of regularization, enhancing alignment with the desired properties as the diffusion process progresses.

## 4 EXPERIMENTS

Considering the practical demands of the pharmaceutical industry, we implement DECOMPDPO to address two critical needs: (1) fine-tuning the reference model for molecule generation across various protein families, and (2) optimizing the reference model specifically for targeted protein subpockets.

### 4.1 EXPERIMENTAL SETUP

**Dataset** We followed prior work (Luo et al., 2021; Peng et al., 2022; Guan et al., 2023a;b) in using the CrossDocked2020 dataset (Francoeur et al., 2020) to pre-train our base model and evaluate the performance of DECOMPDPO. According to the protocol established by Luo et al. (2021), we filtered complexes to retain only those with high-quality docking poses (RMSD $< 1$Å) and diverse protein sequences (sequence identity $< 30\%$), resulting in a refined dataset comprising 100,000 high-quality training complexes and 100 novel proteins for evaluation.

To fine-tune with DECOMPDPO, we sample 10 molecules for each protein in the training dataset using pre-trained base model. The favorability of each molecule was evaluated based on a multi-objective score defined as $r_{\text{multi}} = \sum_{x_i \in X} x_i$, where $X$ denotes the set of normalized optimization objectives. For each protein, we select the molecules with the highest and lowest scores to form preference pairs for the fine-tuning process, resulting in 63,092 valid pairs. For molecular optimization, we sample 500 molecules for each target protein with fine-tuned model in the test dataset and and construct preference pairs from the top 100 and bottom 100 molecules based on their scores.

**Baselines** To assess the capability of DECOMPDPO fine-tuned model in generating high-quality molecules across various protein families, we compare it with several representative generative models. **liGAN** (Ragoza et al., 2022) employs a CNN-based variational autoencoder to encode both ligand and receptor into a latent space, subsequently generating atomic densities for ligands. Atom-based autoregressive models such as **AR** (Luo et al., 2021), **Pocket2Mol** (Peng et al., 2022), and **GraphBP** (Liu et al., 2022) update atom embeddings using a graph neural network (GNN). **TargetDiff** (Guan et al., 2023a) and **DecompDiff** (Guan et al., 2023b) utilize GNN-based diffusion models, the latter innovatively incorporates decomposed priors for predicting atoms' type, position, and bonds with validity guidance. **IPDiff** (Huang et al., 2023) integrates the interactions between pockets and ligands into both the diffusion forward and sampling processes. In addition, to evaluate the molecular optimization capabilities of DECOMPDPO, we compared it with two strong optimization methods. **RGA** (Fu et al., 2022), which utilizes a reinforced genetic algorithm to simulate evolutionary processes and optimize a policy network across iterations, and **DecompOpt** (Zhou et al., 2024a), which leverages decomposed priors to control and optimize conditions in a diffusion model.

**Evaluation** Following the methodology outlined by Guan et al. (2023a), we evaluate molecules from two aspects: **target binding affinity and molecular properties**, and **molecular conformation**. Following the established protocol from previous studies (Luo et al., 2021; Ragoza et al., 2022), we use AutoDock Vina to assess **target binding affinity**. *Vina Score* quantifies the direct binding

Table 1: Summary of different properties of reference molecules and molecules generated by DE-COMPDPO and other generative models. (↑) / (↓) denotes a larger / smaller number is better. Top 2 results are highlighted with **bold text** and underlined text, respectively.

| Methods | Vina Score (↓) | | Vina Min (↓) | | Vina Dock (↓) | | High Affinity (↑) | | QED (↑) | | SA (↑) | | Diversity (↑) | | Success Rate (↑) |
|---|---|---|---|---|---|---|---|---|---|---|---|---|---|---|---|
| | Avg. | Med. | Avg. | Med. | Avg. | Med. | Avg. | Med. | Avg. | Med. | Avg. | Med. | Avg. | Med. | |
| Reference | -6.36 | -6.46 | -6.71 | -6.49 | -7.45 | -7.26 | - | - | 0.48 | 0.47 | 0.73 | 0.74 | - | - | 25.0% |
| LiGAN | - | - | - | - | -6.33 | -6.20 | 21.1% | 11.1% | 0.39 | 0.39 | 0.59 | 0.57 | 0.66 | 0.67 | 3.9% |
| GraphBP | - | - | - | - | -4.80 | -4.70 | 14.2% | 6.7% | 0.43 | 0.45 | 0.49 | 0.48 | **0.79** | **0.78** | 0.1% |
| AR | -5.75 | -5.64 | -6.18 | -5.88 | -6.75 | -6.62 | 37.9% | 31.0% | 0.51 | 0.50 | 0.63 | 0.63 | 0.70 | 0.70 | 7.1% |
| Pocket2Mol | -5.14 | -4.70 | -6.42 | -5.82 | -7.15 | -6.79 | 48.4% | 51.0% | 0.56 | **0.57** | **0.74** | **0.75** | 0.69 | 0.71 | 24.4% |
| TargetDiff | -5.47 | -6.30 | -6.64 | -6.83 | -7.80 | -7.91 | 58.1% | 59.1% | 0.48 | 0.48 | 0.58 | 0.58 | 0.72 | 0.71 | 10.5% |
| IPDiff | **-6.42** | -7.01 | -7.45 | -7.48 | -8.57 | -8.51 | 69.5% | 75.5% | 0.52 | 0.53 | 0.60 | 0.59 | 0.74 | 0.73 | 17.7% |
| DECOMPDIFF* | -5.96 | -7.05 | -7.60 | -7.88 | -8.88 | -8.88 | 72.3% | 87.0% | 0.45 | 0.43 | 0.60 | 0.60 | 0.60 | 0.60 | 28.0% |
| DECOMPDPO | -6.10 | **-7.22** | **-7.93** | **-8.16** | **-9.26** | **-9.23** | **78.2%** | **95.2%** | 0.48 | 0.45 | 0.64 | 0.64 | 0.62 | 0.62 | **36.2%** |

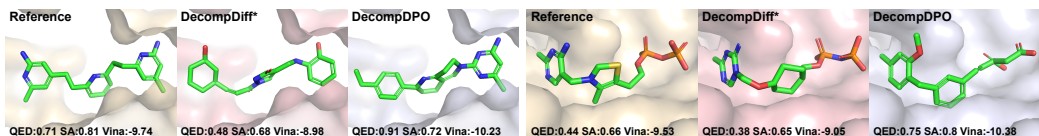

Figure 3: Visualization of reference binding ligands and the molecule generated by DECOMPDIFF* and DECOMPDPO on protein 4D7O (left) and 1UMD (right).

affinity between a molecule and the target protein, *Vina Min* measures the affinity after local structural optimization via force fields, *Vina Dock* assesses the affinity after re-docking the ligand into the target protein, and *High Affinity* measures the proportion of generated molecules with a *Vina Dock* score higher than that of reference ligands. Regarding **molecular properties**, we calculate drug-likeness (*QED*) (Bickerton et al., 2012), synthetic accessibility (*SA*) (Ertl & Schuffenhauer, 2009), and *diversity*. Following Jin et al. (2020); Xie et al. (2021), the overall quality of generated molecules is evaluated by *Success Rate* (QED > 0.25, SA > 0.59, Vina Dock < -8.18). To evaluate **molecular conformation**, Jensen-Shannon divergence (JSD) is employed to compare the atom distributions of the generated molecules with those of reference ligands. We also evaluate median RMSD and energy difference of rigid fragments and the whole molecule before and after optimizing molecular conformations with Merck Molecular Force Field (MMFF) (Halgren, 1996).

**Implementation Details** The bond-first noise schedule proposed by Peng et al. (2023) effectively addresses the inconsistency between atoms and bonds when using predicted bonds for molecule reconstruction. We adapt this noise schedule for DecompDiff, resulting in an enhanced model that we used as our base model, termed as DecompDiff*. Details about the bond-first noise schedule are provided in Appendix C.2. For multi-objective optimization, the optimization objectives for DECOMPDPO and baseline methods are *QED*, *SA*, and *Vina Minimize Score*. During fine-tuning, we assign a weight of 1 to each objective, and for molecular optimization, we weighted each objective by the reciprocal of the distance from the current objectives' mean to the success rate threshold. Please refer to Appendix C for more details.

### 4.2 MAIN RESULTS

**Molecule Generation** We evaluate the effectiveness of DECOMPDPO in terms of target binding affinity and molecular properties. As shown in Table 1, after a single epoch of fine-tuning with DECOMPDPO, the performance is significantly improved across all metrics, demonstrating the effectiveness of DECOMPDPO in multi-objective optimization. Notably, DECOMPDPO achieves the highest score in *Vina Minimize*, *Vina Dock*, *High Affinity*, and *Success Rate* among all generative methods, and also improves other metrics compared to the base model, indicating its superior ability to generate high-quality molecules across various target proteins. Figure 3 shows reference ligands and molecules generated by DecompDiff* and DECOMPDPO. As shown, molecules generated by DECOMPDPO achieve better performance while maintaining desired molecular conformations. More visualize results are provided in Appendix D.

Regarding molecular conformation, we plot the all-atom pairwise distance distribution of generated molecules and compute the JSD with the distribution obtained from reference ligands. As shown in Figure 4, DECOMPDPO has performance comparable to DecompDiff*, achieving the lowest JSD

Table 2: Summary of different properties of reference molecules and molecules generated by DE-COMPDPO and other optimization methods. (↑) / (↓) denotes a larger / smaller number is better. Top 2 results are highlighted with **bold text** and underlined text, respectively.

| Methods | Vina Score (↓) | | Vina Min (↓) | | Vina Dock (↓) | | High Affinity (↑) | | QED (↑) | | SA (↑) | | Diversity (↑) | | Success Rate (↑) |
|---|---|---|---|---|---|---|---|---|---|---|---|---|---|---|---|
| | Avg. | Med. | Avg. | Med. | Avg. | Med. | Avg. | Med. | Avg. | Med. | Avg. | Med. | Avg. | Med. | |
| RGA | - | - | - | - | -8.01 | -8.17 | 64.4% | 89.3% | **0.57** | **0.57** | 0.71 | 0.73 | 0.41 | 0.41 | 46.2% |
| DecompOpt | -5.87 | -6.81 | -7.35 | -7.72 | -8.98 | -9.01 | 73.5% | 93.3% | 0.48 | 0.45 | 0.65 | 0.65 | 0.60 | 0.61 | **52.5%** |
| DECOMPDPO | **-7.27** | **-7.93** | **-8.91** | **-8.88** | **-9.90** | **-10.08** | **88.5%** | **100.0%** | 0.48 | 0.47 | 0.60 | 0.62 | 0.61 | 0.62 | 52.1% |

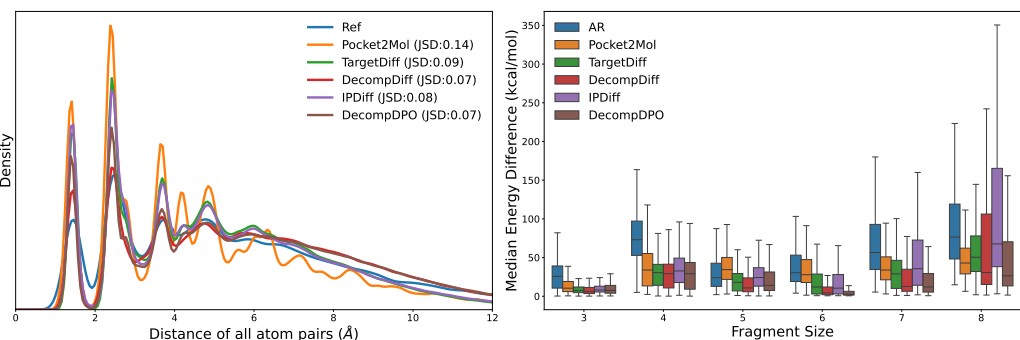

Figure 4: Compare pairwise distance distributions between all atoms in generated molecules and reference molecules from the test set. Jensen-Shannon divergence (JSD) between two distributions is reported (left). Median energy difference for rigid fragments of generated molecules before and after optimizing with the Merck Molecular Force Field (right).

relative to the distribution of reference molecules among all generative models. We also calculate the JSD of bond distance and bond angle distributions, observing that DECOMPDPO does not significantly compromise molecular conformation while achieving superior optimization results. These results are reported in Appendix D. To further evaluate molecular conformation, we calculate the median energy difference before and after conformation optimization by MMFF for rigid fragments that do not contain rotatable bonds. As shown in Figure 4, DECOMPDPO performs comparably to DecompDiff* when with fewer rotatable bonds and achieves the lowest energy differences among all generative methods when with more rotatable bonds. These results demonstrates the potency of DECOMPDPO in maintaining reasonable conformations while optimizing towards desired properties. Results of the median RMSD differences for rigid fragments and whole molecules are provided in Appendix D.

**Molecule Optimization** To validate the capability of DECOMPDPO in molecular optimization, we perform iterative DPO, optimizing the DECOMPDPO fine-tuned model for each target protein in the test set. As shown in Table 2, DECOMPDPO achieves the highest scores in affinity-related metrics among all optimization methods. Compared to DecompOpt, DECOMPDPO achieves a comparable *Success Rate*, demonstrating its effectiveness in continuously enhancing molecule performance toward a specific protein of interest in practical pharmaceutical applications. Additionally, DECOMPDPO can be adapted to DecompOpt, potentially providing stronger results by combining the benefits of preference alignment with iterative optimization.

### 4.3 ABLATION STUDIES

**Single-Objective Optimization** To further validate the effectiveness of DECOMPDPO, we test its performance in single-objective optimization. As AliDiff (Gu et al., 2024) aims to align diffusion models with high binding affinity, we select *Vina Minimize* as the optimization objective and compare the performance of DECOMPDPO with AliDiff. As shown in Table 3, DECOMPDPO achieves higher *Vina Minimize*, *Vina Dock*, *High Affinity*, and *SA* compared to AliDiff. Notably, while only optimizing towards high *Vina Minimize*, DECOMPDPO attains remarkable improvements in molecular properties compared to the base model. Specifically, DECOMPDPO achieves 5.9%, 11.7%, and 11.8% improvement in *Vina Score*, *Vina Minimize*, and *Vina Dock*, respectively, and improvements of 6.7% and 10% in *QED* and *SA*. Additionally, we computed the *Complete Rate*, defined as the percentage of valid and connected molecules among all generated molecules. The *Complete Rate* of DECOMPDPO is 83.6%, representing a 14.8% improvement compared to the base model. These results indicate

Table 3: Summary of results of single-objective optimization for affinity-related metrics. (↑) / (↓) denotes a larger / smaller number is better. The best result is highlighted with **bold text**.

| Method | Vina Score (↓) | | Vina Min (↓) | | Vina Dock (↓) | | High Affinity (↑) | | QED (↑) | | SA (↑) | | Diversity (↑) | | Success |
| | Avg. | Med. | Avg. | Med. | Avg. | Med. | Avg. | Med. | Avg. | Med. | Avg. | Med. | Avg. | Med. | Rate (↑) |
| --- | --- | --- | --- | --- | --- | --- | --- | --- | --- | --- | --- | --- | --- | --- | --- |
| AliDiff | **-7.07** | **-7.95** | -8.09 | -8.17 | -8.90 | -8.81 | 73.4% | 81.4% | **0.50** | **0.50** | 0.57 | 0.56 | **0.73** | **0.71** | - |
| DECOMPDPO | -6.31 | -7.70 | **-8.49** | **-8.72** | **-9.93** | **-9.77** | **85.9%** | **97.8%** | 0.48 | 0.46 | **0.66** | **0.66** | 0.65 | 0.65 | 43.0% |

Table 4: Ablation study of decomposing DPO loss and linear beta schedule. (↑) / (↓) denotes a larger / smaller number is better. The best result is highlighted with **bold text**.

| Method | Vina Score (↓) | | Vina Min (↓) | | Vina Dock (↓) | | High Affinity (↑) | | QED (↑) | | SA (↑) | | Diversity (↑) | | Success |
| | Avg. | Med. | Avg. | Med. | Avg. | Med. | Avg. | Med. | Avg. | Med. | Avg. | Med. | Avg. | Med. | Rate (↑) |
| --- | --- | --- | --- | --- | --- | --- | --- | --- | --- | --- | --- | --- | --- | --- | --- |
| w/ Constant Beta Weight | -5.97 | -7.14 | -7.78 | -8.04 | -9.04 | -9.09 | 74.9% | 91.8% | 0.46 | 0.44 | 0.62 | 0.62 | 0.61 | 0.61 | 32.1% |
| w/ Molecule-level DPO | -6.08 | -7.21 | -7.92 | -8.16 | -9.06 | -9.20 | 77.8% | **96.2%** | **0.48** | **0.45** | 0.63 | 0.63 | 0.60 | 0.61 | 35.1% |
| DECOMPDPO | **-6.10** | **-7.22** | **-7.93** | **-8.16** | **-9.26** | **-9.23** | **78.2%** | 95.2% | **0.48** | **0.45** | **0.64** | **0.64** | 0.62 | 0.62 | **36.2%** |

that DECOMPDPO not only enhances the targeted optimization objective but also improves overall molecular quality and validity, demonstrating its effectiveness in single-objective optimization.

**Benefits of Decomposed Preference**  Our primary hypothesis is that introducing decomposition into the optimization objectives enhances training efficiency by providing greater flexibility in preference selection and multi-objective optimization. We verify this hypothesis in the molecule generation setting by fine-tuning the base model with molecule-level preference pairs for all optimization objectives, which we term Molecule-level DPO. As shown in Table 4, DECOMPDPO outperforms Molecule-level DPO on most of the affinity-related metrics and achieves a higher *Success Rate*, validating that decomposed preference enhances optimization effectiveness and efficiency. Besides, DECOMPDPO achieves a higher *SA*, indicating that it potentially mitigates conflicts in multi-objective optimization by providing greater flexibility and diversity in preference selection.

**Benefits of Linear Beta Schedule**  To validate the effectiveness of the linear beta schedule proposed in Section 3.4, we evaluate the performance of DECOMPDPO when the value of $\beta$ remains constant in molecule generation setting. As shown in Table 4, employing the linear beta schedule improves all metrics, with only a negligible decrease in *Diversity*, indicating its effectiveness in enhancing optimization efficiency.

## 5 CONCLUSION

In this work, we introduced preference alignment to SBDD for the first time, developing DECOMPDPO to align pre-trained diffusion models with multi-granularity preference, which provides more flexibility during the optimization process. The physics-informed energy term penalizing the reward is beneficial for maintaining reasonable molecular conformations during optimization. The linear beta schedule effectively improves optimization efficiency by progressively reducing regularization during the diffusion process. DECOMPDPO shows promising results in molecule generation and molecular optimization, highlighting its ability to meet practical needs of the pharmaceutical industry.

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

## A    IMPACT STATEMENTS

Our contributions to structure-based drug design have the potential to significantly accelerate the drug discovery process, thereby transforming the pharmaceutical research landscape. Furthermore, the versatility of our approach allows for its application in other domains of computer-aided design, including, but not limited to, protein design, material design, and chip design. While the potential impacts are ample, we underscore the importance of implementing our methods responsibly to prevent misuse and potential harm. Hence, diligent oversight and ethical considerations remain paramount in ensuring the beneficial utilization of our techniques.

## B    LIMITATIONS

While we demonstrate that DECOMPDPO excels in improving models in terms of several prevalently recognized properties of molecules. A more comprehensive optimization objectives properties still require attention. Besides, we simply combined the various objectives into a single one by using a weighted sum loss, without investigating the optimal approach for multi-objective optimization. Extending the applicability of DECOMPDPO to more practical scenarios is reserved for our future work.

## C    IMPLEMENTATION DETAILS

### C.1    FEATURIZATION

Following DecompDiff (Guan et al., 2023b), we characterize each protein atom using a set of features: a one-hot indicator of the element type (H, C, N, O, S, Se), a one-hot indicator of the amino acid type to which the atom belongs, a one-dimensional indicator denoting whether the atom belongs to the backbone, and a one-hot indicator specifying the arm/scaffold region. We define the part of proteins that lies within 10Å of any atom of the ligand as pocket. Similarly, a protein atom is assigned to the arm region if it lies within a 10Å radius of any arm; otherwise, it is categorized under the scaffold region. The ligand atom is characterized with a one-hot indicator of element type (C, N, O, F, P, S, Cl) and a one-hot arm/scaffold indicator. The partition of arms and scaffold is predefined by a decomposition algorithm proposed by DecompDiff.

We use two types of message-passing graphs to model the protein-ligand complex: a $k$-nearest neighbors (knn) graph for all atoms (we choose $k = 32$ in all experiments) and a fully-connected graph for ligand atoms only. In the knn graph, edge features are obtained from the outer product of the distance embedding and the edge type. The distance embedding is calculated using radial basis functions centered at 20 points between 0Å and 10Å. Edge types are represented by a 4-dimensional one-hot vector, categorizing edges as between ligand atoms, protein atoms, ligand-protein atoms or protein-ligand atoms. For the fully-connected ligand graph, edge features include a one-hot bond type indicator (non-bond, single, double, triple, aromatic) and a feature indicating whether the bonded atoms belong to the same arm or scaffold.

### C.2    MODEL DETAILS

Our based model used in DECOMPDPO is the model proposed by Guan et al. (2023b), incorporating the bond first noise schedule presented by Peng et al. (2023). Specifically, the noise schedule is defined as follows:

$$s = \frac{s_T - s_1}{\text{sigmoid}(-w) - \text{sigmoid}(w)}$$
$$b = \frac{s_1 + s_T + s}{2}$$
$$\bar{\alpha}_t = s \cdot \text{sigmoid}(-w(2t/T - 1)) + b$$

For atom types, the parameters of noise schedule are set as $s_1 = 0.9999$, $s_T = 0.0001$, $w = 3$. For bond types, a two-stage noise schedule is employed: in the initial stage ($t \in [1, 600]$), bonds are rapidly diffused with parameters $s_1 = 0.9999$, $s_T = 0.001$, $w = 3$. In the subsequent stage ($t \in [600, 1000]$), the parameters are set as $s_1 = 0.001$, $s_T = 0.0001$, $w = 2$. The schedules of atom and bond type are shown in Figure 5.

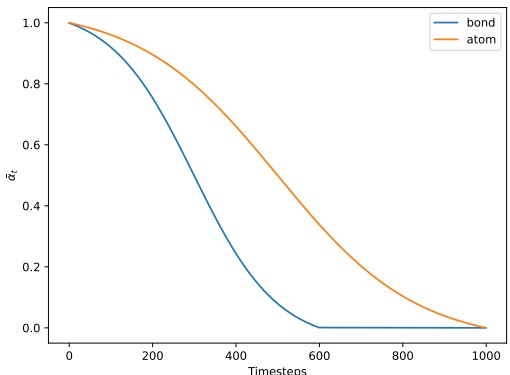

Figure 5: Noise schedule of atom and bond types.

### C.3 MOLECULAR FRAGMENTATION

Following DecompDiff (Guan et al., 2023b), we fragment a molecule into arms and scaffold using RDKit and Alphaspace2 (Katigbak et al., 2020) toolkit. Specifically, subpockets for the target protein is extracted using Alphaspace2 and ligands are decomposed into fragments using BRICS. Then terminal fragments with only one connection site are assigned to subpockets by a linear sum assignment. Arms centers are defined as the centroids of terminal fragments and any remaining subpockets, and scaffold center is defined as the farthest fragment from all arm centers. Finally, the nearest neighbor clustering is performed to tag fragments as arms or the scaffold.

### C.4 TRAINING DETAILS

**Pre-training** We use Adam (Kingma & Ba, 2014) for pre-training, with `init_learning_rate=0.0004` and `betas=(0.95,0.999)`. The learning rate is scheduled to decay exponentially with a factor of 0.6 with `minimize_learning_rate=1e-6`. The learning rate is decayed if there is no improvement for the validation loss in 10 consecutive evaluations. We set `batch_size=8` and `clip_gradient_norm=8`. During training, a small Gaussian noise with a standard deviation of 0.1 to protein atom positions is added as data augmentation. To balance the magnitude of different losses, the reconstruction losses of atom and bond type are multiplied with weights $\gamma_v = 100$ and $\gamma_b = 100$, separately. We perform evaluations for every 2000 training steps. The model is pre-trained on a single NVIDIA A6000 GPU, and it could converge within 21 hours and 170k steps.

**Fine-tuning and Optimizing** For both fine-tuning and optimizing model with DECOMPDPO, we use the Adam optimizer with `init_learning_rate=1e-6` and `betas=(0.95,0.999)`. We maintain a constant learning rate throughout both processes. We set `batch_size=4` and `clip_gradient_norm=8`. Consistent with pre-training, Gaussian noise is added to protein atom positions, and we use a weighted reconstruction loss. For fine-tuning model for molecule generation, we set $\beta_T = 0.001$ and trained for 1 epoch, 16k steps on one NVIDIA A40 GPU. For molecular optimization, we set $\beta_T = 0.02$ and trained for 20,000 steps on one NVIDIA V100 GPU, and perform evaluation every every 1,000 steps.

### C.5 EXPERIMENT DETAILS

The scoring function for selecting training molecules is defined as $S = QED + SA + Vina\_Min/(-12)$. *Vina Minimize Score* is divided by -12 to ensure that it is generally ranges between 0 and 1. For molecule generation, we exclude molecules that cannot be decomposed or reconstructed, resulting in a total of 63,092 preference pairs available for fine-tuning. In molecular optimization, to ensure that the model maintains a desirable completion rate, we include an additional 50 molecules that failed in reconstruction in as the losing side of preference pairs. To tailor the optimization to a specific protein, the weights of the optimization objectives are defined as $w_x = e^{-(x-x_s)}$, where $x$ is the mean property of the generated molecules and $x_s$ is the threshold of the property used in *Success Rate*. For both molecule generation and molecular optimization, we employ the same *Opt Prior* used in DecompDiff. *Opt Prior* is defined as a mixture of *Ref Prior*, which is determined by the reference ligand, and *Pocket Prior*, which is defined by a prior generation algorithm using AlphaSpace2 (Katigbak et al., 2020), depending on whether Ref Prior passes the

*Success threshold.* The $\lambda$ used for penalizing rewards with energy terms proposed in Section 3.3 is set to 0.1.

In evaluating the performance of DECOMPDPO, for each checkpoint, we generate 100 molecules for the molecule generation task and 20 molecules for the molecular optimization task across each target protein in the test set. For both molecule generation and optimization, we select the checkpoint with the highest *weighted Success Rate*, which is defined as the product of the *Success Rate* and the *Complete Rate*.

# D    ADDITIONAL RESULTS

## D.1    FULL EVALUATION RESULTS

**Molecular Conformation**    To provide a more comprehensive evaluation of molecular conformations, we compute the JSD of distances for different types of bonds and angles between molecules from generative models and reference molecules. As shown in Table 5 and Table 6, DECOMPDPO achieves the lowest or second lowest JSD for bond types such as 'C=O', 'C-N', and 'C-O', and for angle types such as 'OPO', 'NCC', and 'CC=O'. For other types of bonds and angles, the JSD generally remains similar to that of DecompDiff, demonstrating that DECOMPDPO generally maintains desirable molecular conformations during preference alignment.

Table 5: Jensen-Shannon Divergence of the bond distance distribution between the generated molecules and the reference molecule by bond type, with a lower value indicating better. "-", "=", and ":" represent single, double, and aromatic bonds, respectively. The top 2 results are highlighted with **bold text** and underlined text.

| Bond | liGAN | GraphBP | AR | Pocket2Mol | TargetDiff | DecompDiff | IPDiff | DECOMPDPO |
|------|-------|---------|-----|-----------|-----------|-----------|--------|-----------|
| C−C | 0.601 | 0.368 | 0.609 | 0.496 | 0.369 | **0.359** | 0.451 | 0.426 |
| C=C | 0.665 | 0.530 | 0.620 | 0.561 | **0.505** | 0.537 | 0.530 | 0.542 |
| C−N | 0.634 | 0.456 | 0.474 | 0.416 | 0.363 | **0.344** | 0.411 | 0.363 |
| C=N | 0.749 | 0.693 | 0.635 | 0.629 | **0.550** | 0.584 | 0.567 | 0.582 |
| C−O | 0.656 | 0.467 | 0.492 | 0.454 | 0.421 | **0.376** | 0.489 | 0.397 |
| C=O | 0.661 | 0.471 | 0.558 | 0.516 | 0.461 | 0.374 | 0.431 | **0.370** |
| C:C | 0.497 | 0.407 | 0.451 | 0.416 | 0.263 | 0.251 | **0.221** | 0.287 |
| C:N | 0.638 | 0.689 | 0.552 | 0.487 | **0.235** | 0.269 | 0.255 | 0.267 |

Table 6: Jensen-Shannon Divergence of the bond angle distribution between the generated molecules and the reference molecule by angle type, with a lower value indicating better. The top 2 results are highlighted with **bold text** and underlined text.

| Bond | liGAN | GraphBP | AR | Pocket2Mol | TargetDiff | DecompDiff | IPDiff | DECOMPDPO |
|------|-------|---------|-----|-----------|-----------|-----------|--------|-----------|
| CCC | 0.598 | 0.424 | 0.340 | 0.323 | 0.328 | **0.314** | 0.402 | 0.353 |
| CCO | 0.637 | 0.354 | 0.442 | 0.401 | 0.385 | **0.324** | 0.451 | 0.358 |
| CNC | 0.604 | 0.469 | 0.419 | **0.237** | 0.367 | 0.297 | 0.407 | 0.312 |
| OPO | 0.512 | 0.684 | 0.367 | 0.274 | 0.303 | 0.217 | 0.388 | **0.194** |
| NCC | 0.621 | 0.372 | 0.392 | 0.351 | 0.354 | **0.294** | 0.399 | 0.300 |
| CC=O | 0.636 | 0.377 | 0.476 | 0.353 | 0.356 | **0.259** | 0.363 | 0.278 |
| COC | 0.606 | 0.482 | 0.459 | **0.317** | 0.389 | 0.339 | 0.463 | 0.355 |

We also evaluate the median RMSD of rigid fragments before and after optimizing molecular conformations with MMFF. As shown in Figure 6, DECOMPDPO consistently achieves lower RMSD differences than the base model, DecompDiff, across all fragment sizes. The median RMSD and energy differences for whole molecules are presented in Figure 7 and Figure 8, respectively. Generally, DECOMPDPO achieves comparable or even better results than DecompDiff, indicating that it can generate molecular conformations with low energy while optimizing towards preference. We further provided numerical evidence in addition to the distributional evidence in Table 7.

**Molecular Properties**    To provide a comprehensive evaluation, we have expanded our evaluation metrics beyond those discussed in Section 4.1, which primarily focus on molecular properties and binding affinities. To assess the model's efficacy in designing novel and valid molecules, we calculate the following additional metrics:

Table 7: Summary of conformation related metrics of generated molecules. RF is short for rigid fragments. The top 2 results are highlighted with **bold text** and underlined text.

|  | Pocket2Mol | TargetDiff | IPDiff | DecompDiff | DecompDPO |
|---|---|---|---|---|---|
| JSD - All Atom | 0.14 | 0.09 | 0.08 | 0.07 | 0.07 |
| Energy Diff - RF | 31.18 | 1355.94 | 1459.45 | 39.39 | 42.49 |
| Energy Diff - Mol | 185.14 | 6116.37 | 21431.71 | 8833.80 | 976.33 |
| RMSD - RF | 0.12 | 0.13 | 0.14 | 0.13 | 0.11 |
| RMSD - Mol | 0.75 | 1.02 | 1.04 | 1.10 | 1.11 |

- *Complete Rate* is the percentage of generated molecules that are connected and vaild, which is defined by RDKit.

- *Novelty* is defined as the ratio of generated molecules that are different from the reference ligand of the corresponding pocket in the test set.

- *Similarity* is the Tanimoto Similarity between generated molecules and the corresponding reference ligand.

- *Uniqueness* is the proportion of unique molecules among generated molecules.

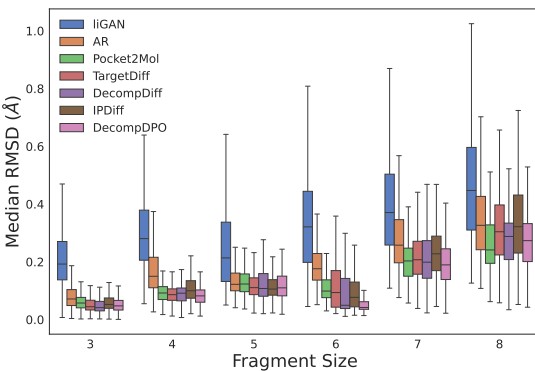

Figure 6: Median RMSD for rigid fragments of generated molecules before and after optimizing with the Merck Molecular Force Field

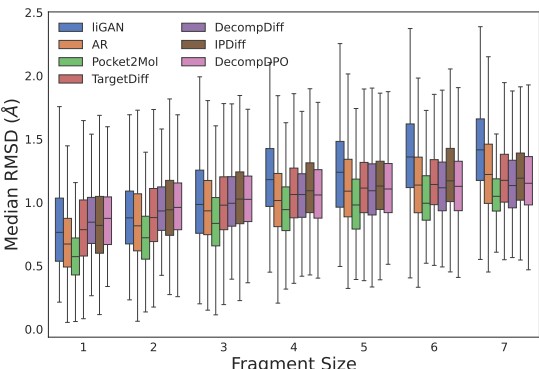

Figure 7: Median RMSD of generated molecules before and after optimizing with the Merck Molecular Force Field

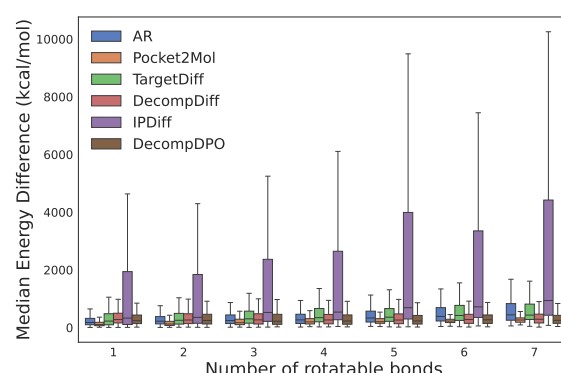

Figure 8: Median energy difference of generated molecules before and after optimizing with the Merck Molecular Force Field

As reported in Table 8, in molecule generation, DECOMPDPO fine-tuned model achieves better *Complete Rate* and *Similarity* compared to the base model. In molecular optimization, DECOMPDPO maintains a relatively acceptable *Complete Rate* and the lowest similarity among all optimization methods.

Table 8: Summary of the models' ability in designing novel and valid molecules. ($\uparrow$) / ($\downarrow$) denotes a larger / smaller number is better.

|  | Methods | Complete Rate ($\uparrow$) | Novelty ($\uparrow$) | Similarity ($\downarrow$) | Uniqueness ($\uparrow$) |
|---|---|---|---|---|---|
| Generate | LiGAN | 99.11% | 100% | 0.22 | 87.82% |
|  | AR | 92.95% | 100% | 0.24 | 100% |
|  | Pocket2Mol | 98.31% | 100% | 0.26 | 100% |
|  | TargetDiff | 90.36% | 100% | 0.30 | 99.63% |
|  | DECOMPDIFF* | 72.82% | 100% | 0.27 | 99.58% |
|  | DECOMPDPO | 73.26% | 100% | 0.26 | 99.57% |
| Optimize | RGA | - | 100% | 0.37 | 96.82% |
|  | DecompOpt | 71.55% | 100% | 0.36 | 100% |
|  | DECOMPDPO | 65.05% | 100% | 0.26 | 99.63% |

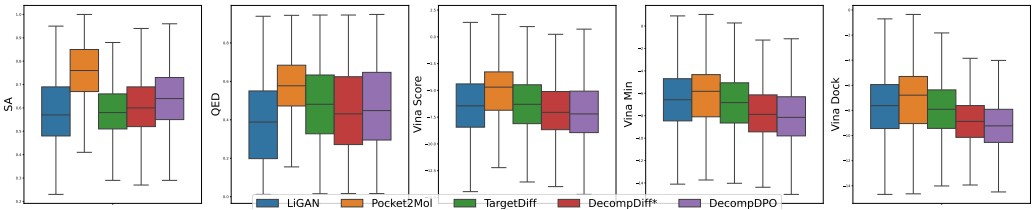

Figure 9: Boxplots of QED, SA, Vina Score, Vina Minimize, and Vina Dock of molecules generated by DECOMPDPO and other generative models.

We also draw boxplots to provide confidence intervals for the performance in molecule generation, which are shown in Figure 9.

To further illustrate the potency of the generated molecules, we draw a scatter plot of heavy atom numbers versus Vina Dock score to demonstrate the effect of heavy atom numbers on the binding affinity of generated molecules.

## D.2 EVIDENCE FOR DECOMPOSABILITY OF PROPERTIES

As illustrated in Section 3.2, *QED* and *SA* are non-decomposable due to the non-linear processes involved in their calculations. We validate this non-decomposability on our training set. As shown in Figure 11, the Pearson correlation coefficients between the properties of molecules and the sum of the properties of their decomposed substructures are very low, not exceeding 0.1. These results indicate that substructures with higher *QED* or *SA* do not necessarily lead the molecule to have better properties. Therefore, we choose molecule-level preferences for *QED* and *SA*.

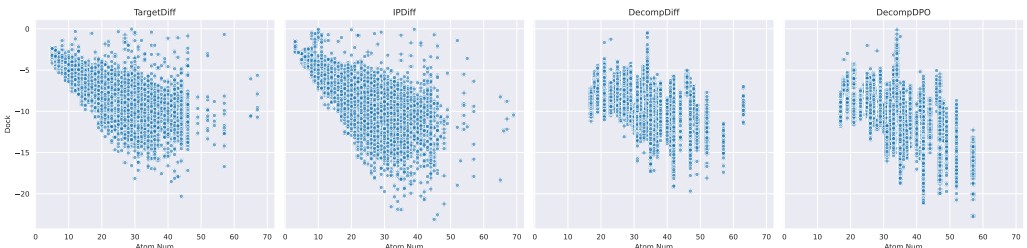

Figure 10: Scatter Plots of heavy atom numbers versus Vina Dock scores for TargetDiff, IPDiff, DecompDiff, and DECOMPDPO.

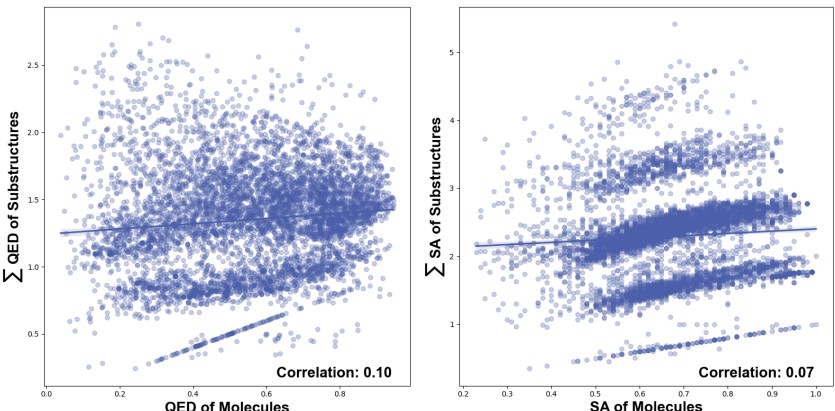

Figure 11: The Pearson correlation between molecule's and sum of substructure's SA (left) / QED (right) Scores in the training dataset.

### D.3 TRADE-OFF IN MULTI-OBJECTIVE OPTIMIZATION

Given the multiple objectives in DECOMPDPO, inherent trade-offs between different properties are unavoidable. In molecular optimization, as illustrated in Figure 12, molecules generated by DECOMPDPO exhibit significantly improved properties compared to those generated by DecompDiff. However, DECOMPDPO encounters a notable trade-off between optimizing the *Vina Minimize Score* and *SA*.

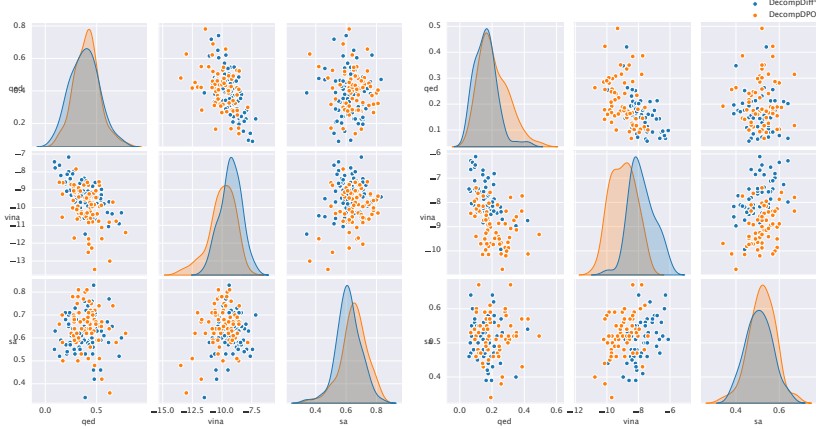

Figure 12: Pairplots of molecules' properties before and after using DECOMPDPO for molecular optimization on protein 4Z2G (left) and 2HCJ (right).

### D.4 TRAINING SET DISTRIBUTION

We further provided winning and losing molecules' distribution of QED, SA, and Vina Minimize in the training set, as shown in Figure 13.

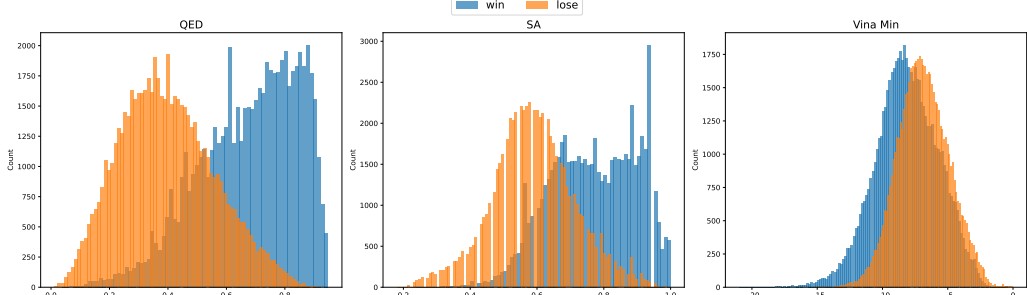

Figure 13: Distribution of QED, SA, and Vina Minimize of the winning and losing molecule in the training set.

### D.5 EXAMPLES OF GENERATED MOLECULES

Examples of reference ligands and molecules generated by DecompDiff* and DECOMPDPO, which are shown in Figure 14.

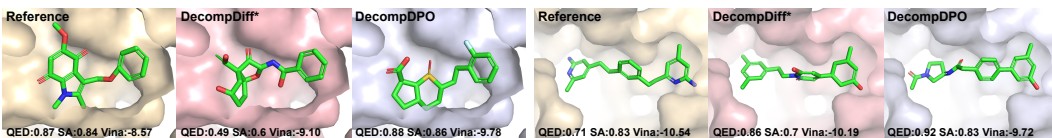

Figure 14: Additional Examples of reference binding ligands and the molecule with the highest property among all generated molecules of DECOMPDIFF* and DECOMPDPO on protein 1GG5 (left) and 3TYM (right).

