# OpenReview forum: "Decomposed Direct Preference Optimization for Structure-Based Drug Design"
_ICLR.cc/2025/Conference — Submitted to ICLR 2025_

### Official Review · Reviewer_koEE · 2024-10-17

**Soundness:** 2
**Presentation:** 2
**Contribution:** 1
**Rating:** 3
**Confidence:** 5

**Summary:**

DecompDPO introduces substructure decomposition into the Direct Preference Optimization of a diffusion model for structure-based drug design. It has two primary uses: fine-tuning diffusion models across various protein families and secondary optimization post-generation. DecompDPO also directly aligns itself with preferences using two forms of DPO: Global over molecule pairs and Local over substructures. DecompDPO was benchmarked on existing CrossDocked2020 benchmarks.

**Strengths:**

Overall, DecompDPO shows a well-done approach to taking complex decomposed molecule diffusion models and optimizing them for easy-to-calculate chemical properties. Many components of the DPO optimization are specific to the way DecompDiff frames the arms and scaffolds of molecule generation. The work in Appendix D.2 is really interesting and would suggest being a primary focus of the paper as this is where the most novelty as it pertains the DPO setup is.

- DecompDPO shows 8% improvement over base model DecompDiff due to DPO fine-tuning with a single epoch of  63K pairs with a pretraining dataset of 100K.
- DecompDPO brings in several innovations from prior methods, including DecompDiff and the bond-first noise schedule, into a single model.
- DecompDPO adds structural queues from bond distances and angles to derive the DPO reward
- From Table 2, it is clear that fine-tuning for multi-objective preferences improves base unaltered predicted structure compared to prior optimization methods on top of DecompDiff.

**Weaknesses:**

Overall, the paper combines many techniques from prior works, but the benchmarking is quite sparse regarding critical ablations. Many architectural and DPO-specific optimization choices are not ablated, although the proposed novelty is such. As a result, it's hard to understand where the method works and why.

- Relevant prior work [1] is not included, which discusses [3] and CrossDocked Benchmark implications and diffusion multi-objective optimization.
- Vina Dock, as it randomizes all torsions to start, has little bearing on the strength of the structure aspect of the generative model.
  - [2] demonstrated that an LLM without 3D information can achieve a success rate of 86.4% and an average Vina Dock of -10.27 kcal/mol by optimizing for docking score directly 25.6x faster than DecompOpt.
  -  The molecular properties like QED and SA are also improved by leveraging a pretrained LLM that only guides Vina Dock, whereas the property-aware optimization of DecompDPO does not yield significant gains.
-  The ground truth CrossDocked2020 data was generated with SMINA docking, and as a result, most of the cross-docked ligands (i.e., non, co-crystal) do not exhibit the desired protein-ligand interaction. As a result, benchmarks such as PoseCheck [3] allow for a better but still imperfect comparison but are not included.
- Given that the structures generated from the AliDiff model have better affinity according to the Vina Score, the fact that DecompDPO structures could be optimized with external force fields or dynamic torsion sampling is not all that valuable compared to non-diffusion methods.
   - **Q:** Why does the multi-objective DPO fair worse than the energy-focused version of AliDiff? Could the regressions stem from the IPDiff vs DecompDIff underlying architecture and training? A deeper study of the underlying DPO method agnostic to the architecture would be very interesting as unless one is to use DecompDiff can someone use DecompDPO?
  -  If Vina Dock, QED, and SA score are desired, there are prior methods that are much faster to achieve that [2].
- The complete rate is the lowest of all tested methods, which is the first and most important benchmark as it shows how well the model can function beyond whether the molecules are good or not.
  -  **Q:**  Is there a deeper analysis as to why 35% of the generated molecules are not connected and/or valid?
  - This seems unexpected as the bond diffusion schedule that was used was specifically chosen to generate connected molecules.
  - **Q:**  For evaluation, does this mean that of the 100 molecules generated, the benchmarks are calculated over the 65 valid molecules?
- A lot of focus is put on adding energy terms to improve the molecule's conformation, but DecompDPO has worse bond distances than DecompDiff for 5/8 bond types and 6/7 bond angles.
  - **Q:**  Does this part of the reward add anything of value if it hurts the generated structures?
- Overall, it is clear DecompDPO is better than DecompOpt, but more ablations are needed to understand where the improvements and regressions are coming from.
  - **Q:**  Could a similar DPO scheme be applied to TargetDiff, and would similar regressions occur?

[1] PILOT: Equivariant diffusion for pocket conditioned de novo ligand generation with multi-objective guidance via importance sampling https://arxiv.org/abs/2405.14925

[2] EvoSBDD: Latent Evolution for Accurate and Efficient Structure-Based Drug Design https://openreview.net/pdf?id=sLhUNz0uTz

[3] PoseCheck: Benchmarking Generated Poses: How Rational is Structure-based Drug Design with Generative Models? https://arxiv.org/abs/2308.07413

**Questions:**

For most questions, see weaknesses.

- DecomDPO performs iterative DPO, optimizing the model for each target protein in the test set. Does this mean a new model is trained for each test set pocket? If so, what happens if you only do standard DPO fine-tuning as done in AliDiff?

- On line 350 it appears to describe that DecompDPO samples 500 molecules per 100 test pocket for table 1. Do the benchmarks reflect the average over all 500 or is it some filtered subset?

- What is the inference time, and how does it compare to prior methods factoring in the complete rate?

- What happens if you fine-tune TargetDiff on the DPO fine-tuning dataset as done in AliDiff? It would be interesting to see of the decomposed objective yields better training data.

- Where are results showing fine-tuning pretrained diffusion models for molecule generation across various protein families (line 21-22)? There is no notion of protein families discussed in existing CrossDocked2020 benchmarks.

Nit: Line 77 says DecompDPO was the first, but line 80 says AliDiff was the first to use preference alignment.

---

> ### Author Response · Authors · 2024-11-26
> **Answer to Weakness Part1**
>
> Thank you for your detailed feedback. Please see below for our responses to the comments.
>
> **Q1: Include relevant prior work PILOT.**
>
> A1: Thank you for your suggestion! We have added discussion of PILOT [1] in Related Work section.
>
> **Q2 & Q4.2: Concern related to Vina Dock.**
>
> A2: We appreciate the opportunity to clarify this aspect. As we illustrated in the paper, DecompDPO is a perference alignment framework that designed for aligning practical pharmaceutical needs using multi-granularity preference. While we demonstrated the remarkable performance of DecomDPO in terms of several prevalently recognized molecular properties, its application scope is not only limited to optimize these properties. Besides, compared to DecompOpt [1] and EvoSBDD [2], which only focus on optimizing model towards a specific target protein, DecompDPO is also applicable for molecule generation, highlighting its generalisability of providing better distribution without additional model adjustment toward each target. To provide a more comprehensive discussion of current search of molecular optimization, we have added EvoSBDD in Related Works section.
> To better illustrate the effectiveness of DecompDPO, we extended the training of our model by an additional epoch, totaling 30,000 fine-tuning steps, which aligns with AliDiff. The results are shown as follows:
>
> |                     | Vina Score |            | Vina Min   |            | Vina Dock   |             | High Affinity |            | QED      |          | SA       |          | Diversity |          | Success Rate |
> | ------------------- | ---------- | ---------- | ---------- | ---------- | ----------- | ----------- | ------------- | ---------- | -------- | -------- | -------- | -------- | --------- | -------- | ------------ |
> |                     | mean       | median     | mean       | median     | mean        | median      | mean          | median     | mean     | median   | mean     | median   | mean      | median   | mean         |
> | AliDiff             | \-7.07     | \-7.95     | \-8.09     | \-8.17     | \-8.90      | \-8.81      | 73.4%         | 81.4%      | 0.50     | 0.50     | 0.57     | 0.56     | **0.73**  | **0.71** | \-           |
> | DecompDPO - 1 epoch | \-6.31     | \-7.70     | \-8.49     | \-8.72     | \-9.93      | \-9.77      | 85.9%         | 97.8%      | **0.48** | **0.46** | 0.66     | 0.66     | 0.65      | 0.65     | 43.0%        |
> | DecompDPO - 2 epoch | **\-7.15** | **\-8.04** | **\-9.18** | **\-9.12** | **\-10.26** | **\-10.66** | **91.6%**     | **100.0%** | 0.47     | 0.44     | **0.67** | **0.67** | 0.69      | 0.69     | 45.4%        |
>
> As the result shows, DecompDPO achieves significantly improved results, with higher binding affinity related metrics compared to AliDiff, after longer training. For general molecular properties, the Complete Rate after two epochs of optimization is 77%, a slight decrease from 83.6% after one epoch. These results demonstrates the potency of DecompDPO in optimizing binding affinity related metrics effectively, including Vina Score, while not sacrifycing much molecular properties.
>
> **Q3: Evaluation using PoseCheck benchmark.**
>
> A3: Thank you for your suggestion. We reported the quartiles of Strain Energy (SE), the average number of Steric Clashes, and the number of key interactions, i.e., Hydrogen Bond Donors (HB Donors), Hydrogen Bond Acceptors (HB Acceptors), van-der Waals contacts (vdWs) and Hydrophobic interactions (Hydrophobic) calculated using PoseCheck.
>
> |            | SE     |        |        | Clash | HB Donors | HB Acceptor | Hydrophobic | vdWs  |
> | ---------- | ------ | ------ | ------ | ----- | --------- | ----------- | ----------- | ----- |
> |            | 25%    | 50%    | 75%    | Avg.  | Avg.      | Avg.        | Avg.        | Avg.  |
> | Reference  | 34     | 107    | 196    | 5.51  | 0.87      | 1.42        | 5.06        | 6.61  |
> | TargetDiff | 369    | 1243   | 13871  | 10.84 | 0.63      | 0.98        | 5.43        | 7.92  |
> | DecompDiff | 161.87 | 354.28 | 802.43 | 15.42 | 0.59      | 1.48        | 7.96        | 11.06 |
> | DecompDPO  | 141.14 | 322.82 | 723.90 | 15.05 | 0.43      | 1.31        | 8.25        | 11.01 |
>
> As the table shows, DecompDPO achieves better Strain Energy compared to TargetDiff and DecompDiff. Admittedly, although DecompDPO performs slightly better than DecompDiff, the mean number of Steric Clashes is still high due to few anomalies beta priors used in sampling. We will address this issue in our future work.

---

> ### Author Response · Authors · 2024-11-26
> **Answer to Weakness Part2**
>
> **Q4: Why does the multi-objective DPO fair worse than the energy-focused version of AliDiff? Could the regressions stem from the IPDiff vs DecompDIff underlying architecture and training? A deeper study of the underlying DPO method agnostic to the architecture would be very interesting as unless one is to use DecompDiff can someone use DecompDPO?**
>
> A4: Thank you for your question. We respectfully disagree with Reviewer koEE that the multi-objective DecompDPO is fair worse than the energy-focused version of AliDiff, with QED and SA of AliDiff decreased compared to IPDiff in binding affinity optimization, while multi-objective DecompDPO achieves improvement across all properties.
>
> In multi-ojbective optimization, the gradient of different objectives can conflict with each other, leading to less effective performance on each objective compared to single objective optimization. So we think it is more fair to compare DecompDPO with AliDiff in terms of binding affinity optimization. Admittedly, the performance of preference alignment could be affected by the base model. However, as we demonstrated in previous Q2, with the same length of training steps, DecompDPO achieves better performance in terms of all binding-affinity related metrics compared to AliDiff.
>
> We further conducted an experiment using TargetDiff as base model to answer the question regarding different model architecture. Please refer to Q4 in Answer to Questions.
>
> **Q5: Concerns about the Complete Rate**
>
> A5: Thank you for bringing up this concern. First, we would like to clarify that DecompDPO is evaluated in two settings: In molecule generation, as shown in Table 7 of our paper, DecompDPO achieves a Complete Rate of 73.26%, which is slightly higher than DecompDiff. Besides, as reported in Single-Objective Optimization section, DecompDPO achieves a Complete Rate of 83.6%. These results demonstrate that DecompDPO generally maintains robust performance in generating valid molecules after fine-tuning for different objectives in molecule generation.
>
> Admittedly, in molecular optimization, which optimize the fine-tuned model more aggressively towards each target protein, we observe that the Complete Rate decreases to 65%. As preference alignment push the generated molecules into regions of the chemical space with high rewards that may deviate from the training distribution, in more aggressive optimization, the likelihood of producing invalid or disconnected molecules could be increased. These results suggest that careful consideration is needed to balance trade-offs between optimization objective and general molecular properties during optimization. However, we believe that in molecular optimization, the significant improvements in targeted property optimization could justify the slight 7% reduction in validity.

---

> ### Author Response · Authors · 2024-11-26
> **Answer to Weakness Part3**
>
> **Q6: Concerns about the JS Divergence of bond distance and angle.**
>
> A6: Thank you for your question. Indeed, as we stated in Section 3.3, the purpose of introducing physics-informed energy terms in preference alignment is to maintain reasonable molecular conformations instead of optimizing the geometry of generated molecules. So the bond and angle energy in Formula (11) and (12) is used as constriaint terms to penalizing molecules with high rewards but poor conformations rather than an optimization objective in preference alignment. In Table 5 and 6, we demonstrate that with physically-constrained optimization, DecompDPO generally achieves comparable performance with DecompDiff in terms of molecular conformation, which is align with our motivation. To demonstrate the usefulness of physics-informed energy terms, we have further conducted an experiment without using these terms.
>
> |                                               | Vina Score |        | Vina Min |        | Vina Dock |        | High Affinity |        | QED  |        | SA   |        | Diversity |        | Success Rate |
> | --------------------------------------------- | ---------- | ------ | -------- | ------ | --------- | ------ | ------------- | ------ | ---- | ------ | ---- | ------ | --------- | ------ | ------------ |
> |                                               | mean       | median | mean     | median | mean      | median | mean          | median | mean | median | mean | median | mean      | median | mean         |
> | DecompDPO - w/o physics-informed energy terms | \-6.37     | \-7.26 | \-8.05   | \-8.17 | \-9.41    | \-9.27 | 78.5%         | 94.1%  | 0.48 | 0.46   | 0.63 | 0.63   | 0.61      | 0.62   | 34.2%        |
> | DecompDPO                                     | \-6.10     | \-7.22 | \-7.93   | \-8.16 | \-9.26    | \-9.23 | 78.2%         | 95.2%  | 0.48 | 0.45   | 0.64 | 0.64   | 0.62      | 0.62   | 36.2%        |
>
> |                                               | JSD Dist | Energy Diff - rigid fragments | RMSD - rigid fragments | Energy Diff - Mol | RMSD - Mol |
> | --------------------------------------------- | -------- | ----------------------------- | ---------------------- | ----------------- | ---------- |
> | Pocket2Mol                                    | 0.14     | 31.18                         | 0.12                   | 185.14            | 0.75       |
> | TargetDiff                                    | 0.09     | 1355.94                       | 0.13                   | 6116.37           | 1.02       |
> | IPDiff                                        | 0.08     | 1459.45                       | 0.14                   | 21431.71          | 1.04       |
> | DecompDiff                                    | 0.07     | 39.39                         | 0.13                   | 8833.80           | 1.10       |
> | DecompDPO - w/o physics-informed energy terms | 0.07     | 62.91                         | 0.12                   | 12981.34          | 1.14       |
> | DecompDPO                                     | 0.07     | 42.49                         | 0.11                   | 976.33            | 1.11       |
>
> As the results show, the energy differences for both rigid fragments and generated molecules significantly increase when not using energy terms for penalizing unreasonable conformations. These results suggest that physics-informed energy terms effectively prevents the model from optimizing toward molecules with high rewards but unreasonable conformations without sacrificing much optimization performance.
>
> **Q7: Could a similar DPO scheme be applied to TargetDiff?**
>
> D7: Please refer to the result we provided in Answer to Questions Q4
>
> [1] Cremer, J., Le, T., Noé, F., Clevert, D. A., & Schütt, K. T. (2024). PILOT: Equivariant diffusion for pocket conditioned de novo ligand generation with multi-objective guidance via importance sampling. arXiv preprint arXiv:2405.14925.
>
> [2] Zhou, X., Cheng, X., Yang, Y., Bao, Y., Wang, L., & Gu, Q. (2024). DecompOpt: Controllable and Decomposed Diffusion Models for Structure-based Molecular Optimization. arXiv preprint arXiv:2403.13829.
>
> [3] Reidenbach, D. (2024). EvoSBDD: Latent Evolution for Accurate and Efficient Structure-Based Drug Design. In ICLR 2024 Workshop on Machine Learning for Genomics Explorations.

---

> ### Author Response · Authors · 2024-11-26
> **Answer to Questions Part1**
>
> **Q1: DecomDPO performs iterative DPO, optimizing the model for each target protein in the test set. Does this mean a new model is trained for each test set pocket? If so, what happens if you only do standard DPO fine-tuning as done in AliDiff?**
>
> A1: There appears to be a misunderstanding regarding how DecompDPO is trained and utilized. As we emphasized in our paper, DecompDPO is evaluated in two distinct settings: molecule generation and molecular optimization. In molecule generation setting, we fine-tuned DecompDPO on 63k generated preference pairs from the training data for one epoch, resulting in a single fine-tuned model for evaluation. In molecular optimization setting, we further validated the potential of DecompDPO, optimizing the fine-tuned model specifically for each target protein in the test set. In our paper, except for the results presented in Table 2, all evaluations in our paper are based on a single fine-tuned model.
>
> **Q2: On line 350 it appears to describe that DecompDPO samples 500 molecules per 100 test pocket for table 1. Do the benchmarks reflect the average over all 500 or is it some filtered subset?**
>
> A2: We would like to clarify that from line 346 to 352 in our original paper, we described how we generated preference pairs for DecompDPO instead of how we sampled molecules for evaluation. Specifically, DecompDPO is first fine-tuned using 63,092 valid pairs generated on the training set. The fine-tuned model is used for molecule generation. Then we further optimized the fine-tuned DecompDPO for molecular optimization using preference pairs constructed from 500 molecules sampled for each target protein. For model evaluation, we sampled 100 molecules for each target protein in the test set for both molecule generation and molecular optimization.
>
> **Q3: What is the inference time, and how does it compare to prior methods factoring in the complete rate?**
>
> A3: Thank you for your question. The inference time of DecompDPO depends on the base model DecompDiff used in our experiment. As we demonstrated in Answer to Weakness Q5, the Complete Rate of DecompDPO and DecompDiff are generally on the same level. We test the inference efficiency for generating 20 molecules on average. The inference time for DecompDPO and DecompDiff are 506s.

---

> ### Author Response · Authors · 2024-11-26
> **Answer to Questions Part2**
>
> **Q4: What happens if you fine-tune TargetDiff on the DPO fine-tuning dataset as done in AliDiff? It would be interesting to see of the decomposed objective yields better training data.**
>
> A4: Thank you for your question. We agree that more studies that evaluating DecompDPO on different model architectures would further demonstrate its effectiveness. The GlobalDPO with energy constraints can be easily adapted to general model structures. The LocalDPO requires an additional fragmentation procedure to generate decomposed preferences.
>
> Regarding applying DecompDPO to TargetDiff, DPO expects that perference data is sampled from the optimal distribution and usually starts from the reference model distribution in practice. Sampling preference pairs from TargetDiff’s distribution incurs high computational costs, which we could not accommodate alongside other ablation studies during the rebuttal.
>
> For now, to provide preliminary insights, we conducted a toy experiment using DecompDPO's 63k fine-tuning dataset. Specifically, we first randomly sampled half of the data to fine-tune TargetDiff, and term the fine-tuned model as TargetDiff - ft. This step is to reduce the gap between the reference model and preference data distributions. Then we use the remaining half of the dataset to optimize TargetDiff towards Vina Minimize, resulting in the model termed TargetDiff - dpo.
>
> |                  | Vina Score |        | Vina Min |        | Vina Dock |        | High Affinity |        | QED  |        | SA   |        | Diversity |        | Success Rate |
> | ---------------- | ---------- | ------ | -------- | ------ | --------- | ------ | ------------- | ------ | ---- | ------ | ---- | ------ | --------- | ------ | ------------ |
> |                  | mean       | median | mean     | median | mean      | median | mean          | median | mean | median | mean | median | mean      | median | mean         |
> | TargetDiff       | \-5.47     | \-6.30 | \-6.64   | \-6.83 | \-7.80    | \-7.91 | 58.1%         | 59.1%  | 0.48 | 0.48   | 0.58 | 0.58   | 0.72      | 0.71   | 10.5%        |
> | TargetDiff - ft  | \-4.49     | \-5.15 | \-5.86   | \-6.03 | \-7.35    | \-7.30 | 53.7%         | 50.0%  | 0.50 | 0.51   | 0.60 | 0.59   | 0.73      | 0.72   | 7.6%         |
> | TargetDiff - dpo | \-4.95     | \-5.79 | \-6.33   | \-6.52 | \-7.85    | \-7.86 | 53.1%         | 55.0%  | 0.52 | 0.52   | 0.58 | 0.58   | 0.72      | 0.71   | 9.7%         |
>
> As the result shows, the performance of TargetDiff - ft decreased compared to the original TargetDiff, suggesting that fine-tuning directly on a subset of preference data introduces challenges in aligning TargetDiff with the DecompDiff distribution that generates the preference data. Using a more general data distribution instead of top-tail data could yield better results. Regarding the performance of TargetDiff - dpo, while it is better than TargetDiff - ft, it only partially recovers the performance of the original TargetDiff, indicating that preference alignment introduces potential benefits but requires further refinement for full integration.
>
> **Q5: Where are results showing fine-tuning pretrained diffusion models for molecule generation across various protein families (line 21-22)? There is no notion of protein families discussed in existing CrossDocked2020 benchmarks.**
>
> A5: Thank you for your question. Following Luo et al. [4], only proteins with less than 30% sequence identity are selected in the dataset to ensure protein diversity, which is likely to include a wide range of protein families with different structural and functional characteristics. In the CrossDocked2020 test set, the test proteins such as 3L3N, 4XLI, and 5MGL belong to different protein families. The reference to protein families in our paper is intended to highlight the broad applicability of DecompDPO across different targets and to distinguish the two application scenarios.
>
> [4] Luo, S., Guan, J., Ma, J., & Peng, J. (2021). A 3D generative model for structure-based drug design. Advances in Neural Information Processing Systems, 34, 6229-6239.

---

> > ### Comment · Reviewer_koEE · 2024-11-27
> > **Thank you**
> >
> > Thank you for providing more ablations. A lot of the existing benchmarking exhibits procedural flaws that are not introduced by you. Nonetheless it is important to address these issues as these flaws are quite detrimental to the practical usefulness and theoretical understanding of these methods.
> >
> > 1. Given all prior 3D generative models are unable to generate 100% valid and complete molecules, the impact of validity should be reflected in the benchmarks. Otherwise its impossible to know is the high average due to an overall good model or a few good samples or due to filtering.
> >
> > 2. My concerns with Vina Dock are also still not addressed. The Vina Score rather than the min and dock score are what measure the affinity of the raw output of the model. With this Table 1 shows that DPO finetuning results in a bump of 0.14 kcal/mol compared to base DecompDiff on average which is not normalized based on validity/completeness. The main metrics emphasized in line 424 (Vina Minimize, Vina Dock, High Affinity, and Success Rate) are all a function of the Vina software with Vina Dock being completely independent of the generated 3D structure. https://arxiv.org/pdf/2310.03223 also shows that if the goal is redocking as well as QED and SA 2D based method significantly outperform DecompDiff and DecompOpt. As a result the only meaningful metrics here is Vina score and the structure-based distributional metrics.
> >
> > To this end, in Table 3 AliDiff shows better Vina Score (-7.07 to -6.31), QED, and diversity with DecompDPO having better SA.
> >
> > **Q:** Given Table 3, why does DecompDPO generate worse structures that AliDiff that Vina is able to optimize to predict much higher Vina Dock scores? This is something that would be incredibly valuable to the space as several methods exhibit preference for redocking with generating poor raw structures.
> >
> > 3. A deeper study of the results outside the existing metrics is really warranted given much of these benchmarks exhibit significant flaws that can be taken advantage of especially vina. When compared to a similar DPO method AliDiff, its numbers for the TargetDiff ablations are-5.81 to the above -4.95.
> >
> > **Q:** Why does the DPO training and finetuning hurt TargetDiff here whereas AliDiff's DPO help it?
> >
> > 4. Overall I agree with reviewer HgKr. There seems to be an over reliance on the flawed benchmarks ignoring several critical aspects of model performance. Furthermore methods like TacoGFN (which are not compared against) are faster, able to generate 100% valid molecules, and generate better Vina Dock, QED, and SA. Given DecompDPO is a structure-based methods the ability for the model to generate meaningful structures should be the central focus. If the focus is on the final best case molecular properties, a comparison to TacoGFN is warranted. Lastly, given the above TargetDiff experiments it appears DPO finetuning makes structures worse. Something of which requires a more thorough investigation to understand what the finetuning is doing to the model on a principled level.

---

> ### Author Response · Authors · 2024-11-28
> **Response to Reviewer's Feedback (Part 1)**
>
> Thank you for your prompt feedback. We're glad that our rebuttal has resolved many of your previous questions and concerns. Regarding your remaining concerns, we will address it as follows:
>
> > Re: "Given all prior 3D generative models are unable to generate 100% valid and complete molecules, the impact of validity should be reflected in the benchmarks. Otherwise, it's impossible to know is the high average due to an overall good model or a few good samples or due to filtering."
>
> Thank you for your suggestion. We have moved the validity to the table in the main text.
>
> > Re: "My concerns with Vina Dock are also still not addressed....As a result, the only meaningful metrics here are Vina score and the structure-based distributional metrics. To this end, in Table 3 AliDiff shows better Vina Score (-7.07 to -6.31), QED, and diversity with DecompDPO having better SA."
> > Re: "Given Table 3, why does DecompDPO generate worse structures that AliDiff that Vina is able to optimize to predict much higher Vina Dock scores? This is something that would be incredibly valuable to the space as several methods exhibit preference for redocking with generating poor raw structures."
>
> We would like to kindly remind the reviewer that we have already demonstrated DecompDPO's remarkable performance in optimizing binding affinity with a higher Vina Score, Vina Min, Vina Dock, and High Affinity compared to AliDiff, as demonstrated in the answer to Q2 & Q4.2 in Weakness Part 1. For your convenience, we have reiterated the relevant results below:
>
> |                     | Vina Score |        | Vina Min |        | Vina Dock |         | High Affinity |        | QED  |        | SA   |        | Diversity |        | Success Rate |
> | ------------------- | ---------- | ------ | -------- | ------ | --------- | ------- | ------------- | ------ | ---- | ------ | ---- | ------ | --------- | ------ | ------------ |
> |                     | mean       | median | mean     | median | mean      | median  | mean          | median | mean | median | mean | median | mean      | median | mean         |
> | AliDiff             | \-7.07     | \-7.95 | \-8.09   | \-8.17 | \-8.90    | \-8.81  | 73.4%         | 81.4%  | 0.50 | 0.50   | 0.57 | 0.56   | 0.73      | 0.71   | \-           |
> | DecompDPO - 1 epoch | \-6.31     | \-7.70 | \-8.49   | \-8.72 | \-9.93    | \-9.77  | 85.9%         | 97.8%  | 0.48 | 0.46   | 0.66 | 0.66   | 0.65      | 0.65   | 43.0%        |
> | DecompDPO - 2 epoch | \-7.15     | \-8.04 | \-9.18   | \-9.12 | \-10.26   | \-10.66 | 91.6%         | 100.0% | 0.47 | 0.44   | 0.67 | 0.67   | 0.69      | 0.69   | 45.4%        |
>
> As shown in the table above, after fine-tuning DecompDPO for an additional epoch (a total of 30k steps, matching the fine-tuning steps used in AliDiff), DecompDPO outperforms AliDiff across all binding-related metrics. This demonstrates the effectiveness of DecompDPO in generating molecules with superior structures.
>
> > Re: Why does the DPO training and finetuning hurt TargetDiff here whereas AliDiff's DPO help it?
>
> We would like to clarify that the results we provided in the answer to Q4 are preliminary proof-of-concept experiments aimed at demonstrating the potential of applying our idea to TargetDiff. The key takeaway from this preliminary experiment, is that TargetDiff-DPO outperforms TargetDiff-FT. However, fully implementing TargetDiff-DPO would require a more thorough execution of the idea, which was not feasible within the given time constraints.
>
> **Additionally, we chose to apply our idea to DecompDiff because it outperforms TargetDiff across major metrics and serves as a stronger baseline. We believe that when a stronger baseline is available, it is both reasonable and more impactful to demonstrate the effectiveness of our method using that baseline, as it better highlights the full potential of our approach. Therefore, we do not consider it necessary or fair to be required to apply our method to TargetDiff, which is weaker than DecompDiff.**.

---

> > ### Author Response · Authors · 2024-11-28
> > **Response to Reviewer's Feedback (Part 2)**
> >
> > > Re: "Furthermore methods like TacoGFN (which are not compared against) are faster, able to generate 100% valid molecules, and generate better Vina Dock, QED, and SA. ... a comparison to TacoGFN is warranted."
> >
> > Thank you for your suggestion. We have added discussion of TacoGFN in the Related Work section to provide a more comprehensive review of molecular optimization methods. While we recognize the value of TacoGFN and EvoSBDD, we would like to emphasize that our work, along with the baselines in our paper, focuses on 3D molecular generation. Compared to 1D or 2D generation methods, 3D modeling is inherently more complex and challenging as it involves modeling spatial and geometric information.
> >
> > As shown in Table in EvoSBDD, all 3D generation methods generally perform worse than 1D or 2D generation methods on QED, SA, and binding related metrics. However, this does not diminish the value of 3D modeling and generation. On the contrary, 3D generation methods are essential for structure-based drug design (SBDD), where spatial modeling plays a crucial role in evaluating and optimizing binding affinity to target proteins. Given the distinct objectives and challenges of 3D generation compared to 1D/2D generation methods, we believe that direct comparisons between the two are neither fair nor meaningful.
> >
> > Thank you once again for your thoughtful feedback. Please let us know if you have any further questions and suggestions.

---

> ### Author Response · Authors · 2024-12-02
> **Gentle Reminder**
>
> Dear Reviewer koEE,
>
> Thank you for your valuable feedback and the time you've invested in reviewing our work.
>
> As we near the end of the discussion phase, we wanted to follow up to see if you've could review our responses to your comments. If there are any further concerns we can address, please let us know.
>
> We look forward to your feedback and are happy to address any further questions.
>
> Sincerely,
>
> The Authors

---

### Official Review · Reviewer_HgKr · 2024-10-28

**Soundness:** 2
**Presentation:** 3
**Contribution:** 3
**Rating:** 5
**Confidence:** 4

**Summary:**

This paper presents a new diffusion model for structure-based drug design that improves molecular properties (docking score, drug-likeness score, and synthesizability score) of a base model using Direct Preference Optimization (DPO). In addition to straightforward application of diffusion-DPO on the molecular level, the authors propose to apply a more fine-grained version at the fragment level to further improve their results.
The method's performance is assessed in a multi-objective fine-tuning experiment as well as a target-specific optimization setup.

**Strengths:**

### Motivation & ideation

I believe user preference-guided optimization of generated molecules is an important frontier in structure-based drug design. The presented work is a promising extension of the capabilities of recent machine learning models for this task. It builds on various ideas established in prior works on decomposed diffusion models [1], decomposed molecule optimization [2], and preference optimization for molecules [3], and is a sensible extension thereof. Applying the diffusion DPO loss at a substructure level for decomposable objectives makes sense and likely leads to more efficient training. Some benefits of this approach are also shown empirically in this study (Table 4).
Furthermore, to the best of my knowledge the paper represents one of the first applications of DPO to small molecule design (together with Ref. [4]) and thus provides a novel view on the drug design problem.


### Experimental setup

The chosen experimental setup looks reasonable to me and includes useful comparisons and baselines.
It also already contains most relevant ablation studies, including local (substructure) DPO vs purely global DPO and the proposed schedule for the regularization parameter $\beta$.
The presented results are consistent with the underlying hypotheses and support the proposed preference alignment strategy for molecular diffusion models.


### References

[1] Guan, Jiaqi, et al. "DecompDiff: diffusion models with decomposed priors for structure-based drug design." arXiv preprint arXiv:2403.07902 (2024).

[2] Zhou, Xiangxin, et al. "DecompOpt: Controllable and Decomposed Diffusion Models for Structure-based Molecular Optimization." arXiv preprint arXiv:2403.13829 (2024).

[3] Zhou, Xiangxin, et al. "Antigen-Specific Antibody Design via Direct Energy-based Preference Optimization." arXiv preprint arXiv:2403.16576 (2024).

[4] Gu, Siyi, et al. "Aligning target-aware molecule diffusion models with exact energy optimization." arXiv preprint arXiv:2407.01648 (2024).

**Weaknesses:**

### Empirical results

While the conceptual arguments for the proposed techniques are strong and well-motivated, the empirical results are currently less convincing. The performance improvement compared to the base model is only moderate in most reported metrics (e.g. QED and SA scores in Table 1). Here, it would be useful to put these results into perspective somehow. The authors could, for example, discuss potential ways to increase the performance gap between base model and optimized model. It would be natural to ask if more DPO fine-tuning iterations/epochs can further increase the optimized scores. If that is not the case, it would at least be interesting to know why.

Moreover, when directly optimizing a limited number of target metrics, it is easy to "break things". The evaluation should therefore include additional sanity checks for the generated molecules. The reported _Complete Rate_ is a good first step but a larger set of diverse metrics and checks (e.g. PoseBusters filters as mentioned in the questions below) would increase my confidence in the findings.

### Baselines

The main benchmark in Table 1 contains a representative set of popular structure-based drug design models. However, I would consider these baselines rather weak in this context as most of them were trained to reproduce the training data distribution whereas DecompDPO directly aims to optimize Vina score, QED and SA. More relevant baselines are only considered in the _Molecule Optimization_ section.
Maybe the authors could consider including the same or similar baselines methods in Table 1 as well.

Given the arguably low bar, I find it surprising that DecompDPO does not show larger improvements compared to the presented baseline methods. To give a concrete example, by simply matching the SA values from the dataset (_Reference_) Pocket2Mol seems to achieve better SA scores than DecompDPO which includes SA in its preference alignment scheme.


### Ablation studies

The paper already investigates the impact of many design choices. However, I would also like to see how longer DPO training affects the results. It is currently unclear why the models are only fine-tuned for one epoch. Would longer training improve results or only degrade other molecular properties?
The paper also introduces a physics-inspired constraint on bond and angle geometries and makes claims such as "The physics-informed energy term penalizing the reward is beneficial for maintaining reasonable molecular conformations during optimization" (Conclusion section), but no empirical evidence is provided.

The ablation of the multi-objective optimization approach produced another unexpected result. QED and SA values in the single-objective case (only optimizing for Vina score) are as good or even better than in the multi-objective setup.
It would be good to see an attempt at explaining this outcome and/or a discussion of the implications for the multi-objective approach.

### Discussion

Given some of the surprising results stated above, I believe the paper would benefit from a more in-depth discussion of its empirical findings. It would certainly help to better understand the current limitations of the method.

**Questions:**

- Why is the model only fine-tuned for one epoch? How does longer training affect the target metrics and other molecular properties?

- Is there a trade-off between the target metrics and other molecular properties? Does the general molecular quality degrade when the reward is optimized more aggressively? The PoseBusters [5] filters are widely accepted criteria to perform general sanity checks for molecules, and could be used to answer this question.

- What molecular fragmentation method is used? I could not find this important methodological detail in the paper.

- I would advise to add an ablation study to systematically evaluate the contribution of the physics-informed reward term.

- Optionally, a more extensive evaluation could also include methods like REINVENT [6] or other reinforcement learning/GFlowNets-based approaches that typically do not model molecules explicitly in 3D but can still optimize binding oracles implicitly and are usually strong baselines for molecule optimization tasks.



### Minor comments
- Figure 1: all symbols should be explained in the Figure legend
- lines 177-178: all symbols need to be defined ($\mu_{1:K}$, $\Sigma_{1:K}$, $H$); it would also be useful to describe the data-dependent priors in more detail
- Section 3.1: this section introduces loss components $L_t$ per time step but, in my opinion, should also provide the overall loss function that involves sampling and aggregation of time steps
- Figure 2: font size of axis labels should be increased
- Table 1: in my opinion, the most interesting comparison is the one between the base model, DecompDiff*, and the fine-tuned model to assess the effect of the proposed preference alignment approach. It would be easier to compare the metrics if these two models were presented right below one another in the table.
- When discussing baselines in Section 4.1, it would be interesting to know what rewards RGA and DecompOpt are optimizing for. This would help put the results discussed later into perspective.
- Line 472: the referenced table should be _Table 3_
- Equation 9: sign$(x>y)$ is quite unconventional. I recommend to use sign$(x-y)$ instead
- Equations 8 and 9: the definition of $M_t^+$ and $M_t^-$ as the union of $M^{(i)}$'s can be moved out of the expectation index


### References

[5] Buttenschoen, Martin, Garrett M. Morris, and Charlotte M. Deane. "PoseBusters: AI-based docking methods fail to generate physically valid poses or generalise to novel sequences." Chemical Science 15.9 (2024): 3130-3139.

[6] Loeffler, Hannes H., et al. "Reinvent 4: Modern AI–driven generative molecule design." Journal of Cheminformatics 16.1 (2024): 20.

---

> ### Author Response · Authors · 2024-11-26
> **Answer to Weakness Part1**
>
> Thank you for your constructive and detailed feedback. Please see below for our responses to the comments.
>
> **Q1: Empirical results: The performance improvement compared to the base model is moderate in terms of QED and SA. Will more DPO fine-tuning iterations/epochs further increase the optimized scores? The evaluation should include additional sanity checks such as Complete Rate and PoseBusters filters.**
>
> A1: Thank you for your comment. First, we want to emphasize that in our evaluation, performance of generative models is evaluated across 100 proteins with 100 molecules generated for each protein, ensuring the observed improvement of DecompDPO is robust in statistics and reflects the improvements in distribution. As shown in Table 1, DecompDPO's effectiveness has been validated by its highest SA across all the diffusion based models. To further illustrate its potential, we have further trained DecompDPO for another epoch. Please refer to Q1 in the Answer to Questions.
>
> Second, we reported the Complete Rate, Novelty, Similarity, and Uniqueness in Table 7 to evaluate the model's capability in designing novel and valid molecules. DecompDPO generally maintained similar Complete Rate compared with DecompDiff. To further check the sanity of generated molecules, we reported the PB-Valid checked by the PoseBusters. Please refer to Q2 in the Answer to Questions.
>
> **Q2: Baselines: 1) In Table 1, DecompDPO is compared with baselines that trained to reproduce the training data distribution. More relevant baselines are only considered in the Molecule Optimization section. 2) DecompDPO does not show larger improvements compared to the presented baseline methods, for example Pocket2Mol seems to achieve better SA than DecompDPO.**
>
> A2: Thank you for bringing up this concern. First, as discussed in our original paper, molecule generation emphasizes the generalizability of models, aiming to generate desired molecules across various protein families. This is analogous to evaluating the performance of large language models independently of whether they are fine-tuned using reinforcement learning methods. Out of this concern, we compared DecompDPO with generative models designed for molecule generation to assess its broad applicability.
>
> To provide a more comprehensive comparison, we have included additional target-aware generative methods in the binding affinity optimization setting. We have further incorporated a new baseline, KGDiff [1], with binding knowledge guided generation. The results are summarized in the following table:
>
> |            | Vina Score |           | Vina Min  |           | Vina Dock  |            | High Affinity |           | QED      |          | SA       |          | Diversity |          | Success Rate |
> | ---------- | ---------- | --------- | --------- | --------- | ---------- | ---------- | ------------- | --------- | -------- | -------- | -------- | -------- | --------- | -------- | ------------ |
> |            | mean       | median    | mean      | median    | mean       | median     | mean          | median    | mean     | median   | mean     | median   | mean      | median   | mean         |
> | IPDiff     | \-6.42     | \-7.01    | \-7.45    | \-7.48    | \-8.57     | \-8.51     | 69.5%         | 75.5%     | 0.52     | 0.53     | 0.60     | 0.59     | 0.74      | 0.73     | 17.7%        |
> | DecompDiff | \-5.96     | \-7.05    | \-7.60    | \-7.88    | \-8.88     | \-8.88     | 72.3%         | 87.0%     | 0.45     | 0.43     | 0.60     | 0.60     | 0.60      | 0.60     | 28.0%        |
> | KGDiff     | **−8.04**  | **−8.61** | **−8.78** | **−8.85** | −9.43      | −9.43      | 79.2%         | 87.0%     | **0.51** | **0.51** | 0.54     | 0.54     | \-        | \-       | \-           |
> | AliDiff    | \-7.07     | \-7.95    | \-8.09    | \-8.17    | \-8.90     | \-8.81     | 73.4%         | 81.4%     | 0.50     | 0.50     | 0.57     | 0.56     | **0.73**  | **0.71** | \-           |
> | DecompDPO  | \-6.31     | \-7.70    | \-8.49    | \-8.72    | **\-9.93** | **\-9.77** | **85.9%**     | **97.8%** | 0.48     | 0.46     | **0.66** | **0.66** | 0.65      | 0.65     | 43.0%        |
>
> As the table shows, DecompDPO, initialized from DecompDiff, achieves significant improvements on all binding-affinity-related metrics. Compared to AliDiff and KGDiff, DecompDPO attains the highest Vina Dock scores and High Affinity percentages without sacrificing much on QED and SA values, demonstrating its effectiveness in generating molecules with both high binding affinity and desirable drug-like properties.

---

> ### Author Response · Authors · 2024-11-26
> **Answer to Weakness Part2**
>
> Molecular optimization focuses on generating better molecules for each specific target protein by optimizing more aggressively with specific information about the target binding site. We employ molecular optimization to further validate the effectiveness of DecompDPO in addition to molecule generation, demonstrating its versatility in different settings.
>
> Regarding the comparison with Pocket2Mol, we acknowledge that the performance of preference alignment is influenced by the base model, making it challenging for diffusion models to achieve the exceptionally high SA scores of Pocket2Mol. However, we believe that DecompDPO's superior ability is demonstrated by its outstanding performance, achieving the highest Success Rate among all generative models and the highest SA score among diffusion-based methods, ranking second only to Pocket2Mol among all generative methods. This highlights DecompDPO's effectiveness in balancing high binding affinity with favorable SA score within the diffusion model framework.
>
> **Q3: Ablation studies: 1) How longer DPO training affects the results; 2)Provide empirical evidence for the benefits of physics-informed energy terms in maintaining reasonable molecular conformations during optimization; 3) QED and SA values in the single-objective case are as good or even better than in the multi-objective setup.**
>
> A3: Thank you for your detailed review. We have further provided experiment results with longer training steps. Please refer to Q1 in Answer to Questions.
>
> Second, we have further conducted an experiment without using physics-informed energy terms to penalize dpo reward. We summarized the conformation related metrics in the following table:
>
> |                                               | JSD Dist | Energy Diff - rigid fragments | RMSD - rigid fragments | Energy Diff - Mol | RMSD - Mol |
> | --------------------------------------------- | -------- | ----------------------------- | ---------------------- | ----------------- | ---------- |
> | Pocket2Mol                                    | 0.14     | 31.18                         | 0.12                   | 185.14            | 0.75       |
> | TargetDiff                                    | 0.09     | 1355.94                       | 0.13                   | 6116.37           | 1.02       |
> | IPDiff                                        | 0.08     | 1459.45                       | 0.14                   | 21431.71          | 1.04       |
> | DecompDiff                                    | 0.07     | 39.39                         | 0.13                   | 8833.80           | 1.10       |
> | DecompDPO - w/o energy terms | 0.07     | 62.91                         | 0.12                   | 12981.34          | 1.14       |
> | DecompDPO                                     | 0.07     | 42.49                         | 0.11                   | 976.33            | 1.11       |
>
> The results showed significant increases in energy differences for both rigid fragments and generated molecules when the physics-informed terms were omitted. Specifically, the energy difference for generated molecules increased from 976.33 to 12,981.34. These findings indicate that the inclusion of physics-informed energy terms effectively prevents the model from optimizing toward molecules with high rewards but unreasonable conformations, thereby maintaining more realistic molecular structures during optimization. We also evaluated the binding affinity and molecular properties, which are shown in the following table:
>
> |                                               | Vina Score |        | Vina Min |        | Vina Dock |        | High Affinity |        | QED  |        | SA   |        | Diversity |        | Success Rate |
> | --------------------------------------------- | ---------- | ------ | -------- | ------ | --------- | ------ | ------------- | ------ | ---- | ------ | ---- | ------ | --------- | ------ | ------------ |
> |                                               | mean       | median | mean     | median | mean      | median | mean          | median | mean | median | mean | median | mean      | median | mean         |
> | DecompDPO - w/o energy terms | \-6.37     | \-7.26 | \-8.05   | \-8.17 | \-9.41    | \-9.27 | 78.5%         | 94.1%  | 0.48 | 0.46   | 0.63 | 0.63   | 0.61      | 0.62   | 34.2%        |
> | DecompDPO                                     | \-6.10     | \-7.22 | \-7.93   | \-8.16 | \-9.26    | \-9.23 | 78.2%         | 95.2%  | 0.48 | 0.45   | 0.64 | 0.64   | 0.62      | 0.62   | 36.2%        |
>
> The results indicate that physics-informed energy terms are beneficial in maintaining reasonable molecular conformations without sacrificing much in terms of optimization performance.

---

> ### Author Response · Authors · 2024-11-26
> **Answer to Weakness Part3**
>
> Third, regarding the observed increase in QED and SA, we attribute this to the flexibility introduced by LocalDPO in the decomposed chemical space. To expalin the observed improvements in QED and SA when optimizing for binding affinity, we calculated the Pearson correlation coefficients between the differences in Vina Minimize scores and the differences in QED and SA scores for each pair in our dataset. We found that the correlations between differences in the decomposed arms' Vina Minimize and the corresponding molecules' differences in QED and SA are -0.12 and -0.11, respectively. These statistics suggest that the improvements in QED and SA are potentially due to the benefits of LocalDPO, where the flexibility introduced by the decomposed chemical space alleviates conflicts between SA and binding affinity, allowing the model to navigate the chemical space effectively even when optimizing for binding affinity. Besides, in multi-objective optimization, the gradients of different objectives may conflict with each other, leading to less effective results and potentially hindering the optimization towards QED and SA. Admittedly, in DecompDPO, we do not investigate the optimal approach for multi-objective optimization, which is reserved for our future work.
>
> [1] Qian, H., Huang, W., Tu, S., & Xu, L. (2024). KGDiff: towards explainable target-aware molecule generation with knowledge guidance. Briefings in Bioinformatics, 25(1), bbad435.

---

> ### Author Response · Authors · 2024-11-26
> **Answer to Question Part1**
>
> **Q1: Why is the model only fine-tuned for one epoch? How does longer training affect the target metrics and other molecular properties?**
>
> A1: Thank you for your question regarding the number of fine-tuning epochs. We initially fine-tuned the model for one epoch (15,000 steps) to balance computational efficiency with performance. To assess the impact of longer training, we extended the fine-tuning to two epochs (totaling 30,000 steps), aligning with the fine-tuning steps of AliDiff for a fair comparison. The results are as follows:
>
> |                     | Vina Score |            | Vina Min   |            | Vina Dock   |             | High Affinity |            | QED      |          | SA       |          | Diversity |          | Success Rate |
> | ------------------- | ---------- | ---------- | ---------- | ---------- | ----------- | ----------- | ------------- | ---------- | -------- | -------- | -------- | -------- | --------- | -------- | ------------ |
> |                     | mean       | median     | mean       | median     | mean        | median      | mean          | median     | mean     | median   | mean     | median   | mean      | median   | mean         |
> | AliDiff             | \-7.07     | \-7.95     | \-8.09     | \-8.17     | \-8.90      | \-8.81      | 73.4%         | 81.4%      | 0.50     | 0.50     | 0.57     | 0.56     | **0.73**  | **0.71** | \-           |
> | DecompDPO - 1 epoch | \-6.31     | \-7.70     | \-8.49     | \-8.72     | \-9.93      | \-9.77      | 85.9%         | 97.8%      | **0.48** | **0.46** | 0.66     | 0.66     | 0.65      | 0.65     | 43.0%        |
> | DecompDPO - 2 epoch | **\-7.15** | **\-8.04** | **\-9.18** | **\-9.12** | **\-10.26** | **\-10.66** | **91.6%**     | **100.0%** | 0.47     | 0.44     | **0.67** | **0.67** | 0.69      | 0.69     | 45.4%        |
>
> As the results demonstrate, extending the fine-tuning significantly enhances the binding affinity-related metrics. The median Vina Dock score improved from -9.77 to -10.66, and the High Affinity increased to 100%. For general molecular properties, the Complete Rate after two epochs of optimization is 77%, a slight decrease from 83.6% after one epoch. There is also a minor decrease in QED when training for longer steps. These findings suggest that careful consideration is needed to balance trade-offs between optimization objectives and general molecular properties. However, the substantial improvements confirm that DecompDPO is highly effective in aligning the model with preferences without sacrificing much on overall molecular quality.
>
> **Q2: Is there a trade-off between the target metrics and other molecular properties? Does the general molecular quality degrade when the reward is optimized more aggressively? Posebuster could be used to answer this question.**
>
> A2: Thank you for your insightful question. There is indeed a trade-off between optimizing target metrics and maintaining general molecular properties. When the reward is optimized more aggressively, such as in our molecular optimization setting where DecompDPO is further fine-tuned for each target protein, we observe significant improvements in target metrics, achieving a 52.1% Success Rate as shown in Table 2. To assess whether general molecular quality degrades under this aggressive optimization, we evaluated the Complete Rate and the percentage of valid molecules verified by PoseBuster [3] (PB-Valid):
>
> |                                    | Complete Rate | PB-Valid |
> | ---------------------------------- | ------------- | -------- |
> | DecompDiff                         | 72.82%        | 54.60%   |
> | DecompDPO                          | 73.26%        | 56.56%   |
> | DecompDPO - Molecular Optimization | 65.05%        | 51.90%    |
>
> As the result shows, both the Complete Rate and PB-Valid decrease in molecular optimization compared to the general fine-tuned DecompDPO and baseline model. These results suggest that while more aggressive optimization improves target metrics, it may compromise general molecular properties. Therefore, the trade-off between optimization objectives and general molecular properties should be handled carefully during optimisation.

---

> ### Author Response · Authors · 2024-11-26
> **Answer to Question Part2**
>
> **Q3: Details of molecular fragmentation**
>
> A3: Thank you for your question. We follow the fragmentation algorithm proposed in DecompDiff [1] to decompose molecules. Specifically, we first use Alphaspace2 [2] to extract target protein subpockets and use BRICS to decompose ligands into fragments. Then, terminal fragments (with only one connection site) are assigned to subpockets by a linear sum assignment. Arms centers are defined as the centroids of terminal fragments and any remaining subpockets, and scaffold center is defined as the farthest fragment from all arm centers. Finally, the nearest neighbor clustering is performed to tag fragments as arms or the scaffold. We have added the details of molecular fragmentation in Appendix.
>
> **Q4: Ablation study on the physics-informed reward term.**
>
> A4: Thank you for your question. Please refer to our response to Q3 in Answer to Weakness.
>
> **Q5: More extensive evaluation could also include methods like REINVENT or other reinforcement learning/GFlowNets-based approaches.**
>
> A5: Thank you for your suggestion! We appreciate the importance of including more baseline methods in molecular optimization. Currently, we are still running the experiment. We are committed to include comparisons with REINVENT in the future version of the paper.
>
> **Q6: Response to Minor Comments**
>
> A6: Thank you for your careful reading and valuable suggestions to improve our manuscript. We have addressed all the minor comments in the revised version.
>
>
>
> [1] Guan, J., Zhou, X., Yang, Y., Bao, Y., Peng, J., Ma, J., ... & Gu, Q. (2024). DecompDiff: diffusion models with decomposed priors for structure-based drug design. arXiv preprint arXiv:2403.07902.
>
> [2] Katigbak, J., Li, H., Rooklin, D., & Zhang, Y. (2020). AlphaSpace 2.0: representing concave biomolecular surfaces using β-clusters. Journal of chemical information and modeling, 60(3), 1494-1508.
>
> [3] Buttenschoen, M., Morris, G. M., & Deane, C. M. (2024). PoseBusters: AI-based docking methods fail to generate physically valid poses or generalise to novel sequences. Chemical Science, 15(9), 3130-3139.

---

> > ### Comment · Reviewer_HgKr · 2024-11-27
> > **Response and follow-up questions (1/2)**
> >
> > I would like to thank the authors for their detailed responses and their efforts to address my concerns.
> > While I appreciate the potential of this paper, I still have substantial reservations about its publication in the current state.
> > I am providing detailed explanations for this conclusion as well as my follow-up questions below.
> >
> > ## Responses
> >
> > ### Weaknesses
> >
> > **Q1 & Q2:** Thanks for your explanation and adding more baselines. In my opinion, the additional sanity checks are convincing enough given the small drop compared to the base model. $\approx 60\%$ valid molecules is arguably still sufficient for most practical applications.
> > Please include the new results in the paper. I would also recommend to include either _complete rate_ or _PB-valid_ directly in the main tables to present this data more transparently.
> >
> > However, my main concern about DecompDPO's performance is not about the statistical significance of the results but rather about absolute differences.
> > As I see it, all the presented performance gains are negligible because some baselines perform as well or better than DecompDPO (e.g. Pocket2Mol on QED and SA) even though they are not directly aiming to optimize these properties whereas DecompDPO is.
> > Even the differences to its own base model, DecompDiff, appear to be too small to make a difference in practice.
> > Unfortunately, the newly added baselines (e.g. KGDiff) do not strenghten the claims of the paper either. Importantly, DecompDPO does not perform better than this baseline in the _Vina Minimize_ metric which it is explicitly optimising for. I consider this the core metric as it aligns best with the training objective.
> >
> > **Q3:** Thank you for providing the additional ablations. I am satisfied with the explanations about the physics-informed energy term.
> > Regarding training time, stopping training after two epochs still seems rather arbitrary to me.
> > If more training consistently leads to better results, why is the model not trained for longer and the stronger empirical results reported in the paper?
> > Are there trade-offs that are currently not discussed in the paper and responses?
> >
> > Furthermore, I do not find the explanation for my question about QED and SA values in single-objective vs. multi-objective setups convincing.
> > The authors say that "the gradients of different objectives may conflict with each other, leading to less effective results and potentially hindering the optimization towards QED and SA".
> > If a model attempts to optimize a certain property explicitly but does so less successfully than a model that completely disregards the same property (apart from the very small correlation the authors mentioned in their rebuttal), this calls into question the entire approach in my opinion.
> > Besides, the statement "the flexibility introduced by the decomposed chemical space [LocalDPO] alleviates conflicts between SA and binding affinity" is unsubstantiated because the results in Table 4, which explicitly compares local and global DPO, show there is effectively no difference of the QED and SA values in the two setups.
> >
> > Please note that I am not trying to say the model is not working.
> > As mentioned in my original review, I very much agree with the conceptual arguments the paper makes, but simply believe they are not sufficiently supported by the empirical results.
> > I strongly believe that the proposed approach has great potential if the performance of DecompDPO is further improved.
> >
> > ### Questions
> >
> > **Q1:** See my response to weakness Q3 above.
> >
> > **Q2:** See my reponse to weaknesses Q1-Q3 above.
> >
> > **Q3:** Thanks for adding this explanation.
> >
> > **Q4:** See my response to weakness Q3 above.
> >
> > **Q5:** Thank you for the effort. Do you think this experiment will finish before the end of the discussion period?
> >
> >
> > **Q6:** Still missing:
> > - Figure 1: all symbols should be explained in the Figure legend
> > - Section 3.1: this section introduces loss components $L_t$ per time step but, in my opinion, should also provide the overall loss function that involves sampling and aggregation of time steps
> > - Figure 2: please also increase the font size of axis tick labels.
> > - Equations 8 and 9: the definition of $M_t^+$ and $M_t^-$ as the union of $M^{(i)}$'s can be moved out of the expectation index
> >
> > It is possible that there are good reasons for not including some of my suggestions, but please provide short explanations in those cases and be honest about the fact that you did not address all of them.
> >
> > Additionally, I have this new minor comment:
> > - The caption of Figure 13 needs to be corrected.

---

> > > ### Comment · Reviewer_HgKr · 2024-11-27
> > > **Response and follow-up questions (2/2)**
> > >
> > > ## Remaining concerns
> > >
> > > The main limitation I still see is that a model that is directly trained to optimise one or several molecular properties should demonstrate large, more convincing performance gains in the empirical studies.
> > > Some of the newly added data has the same issue.
> > > For instance, single objective optimization for QED and SA does not seem to improve these properties more than optimization for other quantities and only slightly compared to the DecompDiff baseline.
> > > Therefore it is unclear whether the improvement is actually achieved thanks to the DPO loss or rather because of the energy term or some other design choices.
> > > Overall, I still have concerns about DecompDPO's effectiveness in practice.
> > >
> > > Furthermore, I believe the authors should discuss the empirical results more honestly.
> > > There are several places in which it seems like the paper and rebuttals focus selectively on the metrics that support their narrative while neglecting surprising and less favorable results.
> > > For example, the response in A2 (of the weaknesses) focuses on Vina Dock rather than Vina Min even though the latter seems the be the more relevant metric.
> > >
> > > Finally, the new data is interesting but it is not yet included in the revised paper. Are the authors planning to do so? This concerns
> > > - DecompDPO trained for 2 epochs
> > > - PoseBusters and PoseCheck metrics
> > > - Ablation of the physics-informed energy terms
> > > - Single objective optimization for QED and SA
> > > - New baselines from A2
> > >
> > >
> > > ### Questions
> > >
> > > - $\beta$ schedule: shouldn't regularization be stronger at the end of the trajectory when molecular features are only refined but the global structure is more or less decided already? The optimization objective probably doesn't depend on small geometric details.
> > > - The DecompDPO model trained for 2 epochs seems to perform better. Why is it not trained even longer?

---

> ### Author Response · Authors · 2024-11-29
> **Response to Reviewer's Feedback (Part 1)**
>
> Thank you for your prompt feedback. We are pleased to hear that our rebuttal has addressed many of your previous questions and concerns. Regarding your remaining concerns, we would like to them as follows:
>
> ### Weaknesses
> **Q1 & Q2:**
>
> > Re: "I would also recommend to include either complete rate or PB-valid directly in the main tables to present this data more transparently."
>
> Thank you for your suggestion. We have moved the Complete Rate to the table in the main text and will update in the revised version of paper.
>
> > Re: "However, my main concern about DecompDPO's performance is not about the statistical significance of the results but rather about absolute differences. As I see it, all the presented performance gains are negligible because some baselines perform as well or better than DecompDPO (e.g. Pocket2Mol on QED and SA) even though they are not directly aiming to optimize these properties whereas DecompDPO is. Even the differences to its own base model, DecompDiff, appear to be too small to make a difference in practice."
>
> We would like to emphasise a few key points to address concerns about the performance of DecompDPO:
>
> While Pocket2Mol achieves higher QED and SA scores, its performance on binding affinity-related metrics is notably lower compared to diffusion-based models. This discrepancy highlights an inherent difference between autoregressive models and diffusion models. Autoregressive models tend to excel in chemical property optimization, such as QED and SA, whereas diffusion models are more adept at generating molecules with higher binding affinities.
>
> Therefore, evaluating model performance solely based on a single metric like QED or SA does not provide a complete picture. We believe that the Success Rate, which reflects the overall balance of desirable properties, is a more meaningful indicator of a model's ability. DecompDPO consistently demonstrates a higher Success Rate compared to baseline models, validating its ability to generate high quality molecules.
>
> Furthermore, we would like to reiterate that DecompDPO achieves improvements across multiple metrics, including QED, SA, and all binding-related metrics. Its ability to balance these properties while demonstrating remarkable performance in binding optimization reinforces its utility in SBDD and highlights DecompDPO's potential in optimizing models to generate better molecules.
>
> > Re: "Unfortunately, the newly added baselines (e.g. KGDiff) do not strenghten the claims of the paper either. Importantly, DecompDPO does not perform better than this baseline in the Vina Minimize metric which it is explicitly optimising for. I consider this the core metric as it aligns best with the training objective."
>
> Thank you for your comment. We would like to kindly remind the reviewer that, as addressed in Q1 of Questions Part 1, the results from DecompDPO fine-tuned for two epochs (a total of 30k steps, matching the fine-tuning steps used in AliDiff and KGDiff) demonstrate that DecompDPO achieves higher performance on Vina Minimize. For your convenience, we reorganize the results below:
>
> |                     | Vina Score |           | Vina Min   |            | Vina Dock   |             | High Affinity |            | QED      |          | SA       |          | Diversity |          | Success Rate |
> | ------------------- | ---------- | --------- | ---------- | ---------- | ----------- | ----------- | ------------- | ---------- | -------- | -------- | -------- | -------- | --------- | -------- | ------------ |
> |                     | mean       | median    | mean       | median     | mean        | median      | mean          | median     | mean     | median   | mean     | median   | mean      | median   | mean         |
> | AliDiff             | \-7.07     | \-7.95    | \-8.09     | \-8.17     | \-8.90      | \-8.81      | 73.4%         | 81.4%      | 0.50     | 0.50     | 0.57     | 0.56     | **0.73**  | **0.71** | \-           |
> | KGDiff              | **−8.04**  | **−8.61** | −8.78      | −8.85      | −9.43       | −9.43       | 79.2%         | 87.0%      | **0.51** | **0.51** | 0.54     | 0.54     | \-        | \-       | \-           |
> | DecompDPO - 1 epoch | \-6.31     | \-7.70    | \-8.49     | \-8.72     | \-9.93      | \-9.77      | 85.9%         | 97.8%      | 0.48     | 0.46     | 0.66     | 0.66     | 0.65      | 0.65     | 43.0%        |
> | DecompDPO - 2 epoch | \-7.15     | \-8.04    | **\-9.18** | **\-9.12** | **\-10.26** | **\-10.66** | **91.6%**     | **100.0%** | 0.47     | 0.44     | **0.67** | **0.67** | 0.69      | 0.69     | 45.4%        |
>
> As the table shows, DecompDPO fine-tuned for two epochs achieves higher Vina Minimize, Vina Dock, and High Affinity compared to AliDiff and KGDiff, confirming the effectiveness of DecompDPO in aligning with the core optimization objective.

---

> ### Author Response · Authors · 2024-11-29
> **Response to Reviewer's Feedback (Part 2)**
>
> **Q3:**
>
> > Re: "Thank you for providing the additional ablations. ... why is the model not trained for longer and the stronger empirical results reported in the paper? Are there trade-offs that are currently not discussed in the paper and responses?"
>
> Thank you for recognizing our efforts in providing additional ablations. Regarding training time, as explained in Q1 in Question Part 1, our current results already demonstrate significant improvements over the base model. Out of the concern of computational efficiency, we believe the current training time is sufficient to validate the strong optimization capabilities of DecompDPO.
>
> Additionally, we have discussed potential trade-offs between stronger performance of optimization objective and other general molecular properties in the answer to Q1 in Question Part 1. We reiterate here for clarity: While extending training to two epochs enhances the optimization objective, there are slight decreases in general molecular properties, with the Complete Rate dropped from 83% to 77%, and the mean QED decreased from 0.48 to 0.47.
>
> > Re: "Furthermore, I do not find the explanation for my question about QED and SA values in single-objective vs. multi-objective setups convincing. The authors say that "the gradients of different objectives may conflict with each other, leading to less effective results and potentially hindering the optimization towards QED and SA". If a model attempts to optimize a certain property explicitly but does so less successfully than a model that completely disregards the same property (apart from the very small correlation the authors mentioned in their rebuttal), this calls into question the entire approach in my opinion. Besides, the statement "the flexibility introduced by the decomposed chemical space [LocalDPO] alleviates conflicts between SA and binding affinity" is unsubstantiated because the results in Table 4, which explicitly compares local and global DPO, show there is effectively no difference of the QED and SA values in the two setups."
>
> Thank you for your detailed feedback. We would like to further address your concerns by emphasizing the following key aspects:
>
> As demonstrated in Q3 of Weakness Part 3, our dataset shows correlations between the optimization objectives: The correlations between the differences in the decomposed arms' Vina Minimize and the corresponding molecules' differences in QED and SA are -0.12 and -0.11, respectively. In contrast, the correlation between differences in molecules' Vina Minimize and SA is 0.27, and between Vina Minimize and QED is -0.14. This shift indicates that decomposition fundamentally changes the way Vina Minimize optimization interacts with QED and SA. In the single-objective optimization of Vina Minimize, the negative correlation introduced by decomposition may indirectly lead to improvements in QED and SA, as observed in our results.
>
> We would like to further clarify that multi-objective optimization remains effective and provides balanced improvements across metrics. It does not prioritize a single metric, but seeks a compromise that improves the overall molecular quality. The results of the two-epoch optimization provide clear evidence: When DecompDPO is optimized more aggressively towards Vina Minimize, QED decreased and became lower compared to multi-objective optimization results. This result demonstrates that more aggressive optimization of Vina Minimize may result in reduced secondary benefits for QED and SA due to the inherent trade-offs between molecular properties.
>
> ### Questions
> **A1 & A2**: Please refer to Weakness Q3 above.
>
> **A3 & A4**: Thank you for recognizing our efforts.

---

> ### Author Response · Authors · 2024-11-29
> **Response to Reviewer's Feedback (Part 3)**
>
> **A5:** Thank you for your question. We will try to provide the result before the end of the discussion period and plan to include REINVENT 4 in the Related Work section to provide a more comprehensive overview.
>
> We would also like to clarify an important distinction: our work, along with the baselines in the paper, focuses on 3D molecular generation, whereas REINVENT 4 operates in a 1D space using SMILES. The challenges and goals of 3D and 1D molecular generation are very different. Compared to 1D or 2D generation methods, 3D modelling involves capturing spatial and geometric information, which is inherently more complex.
>
> As noted in EvoSBDD [1], a 1D optimization method based on SMILES, 3D generation methods generally perform worse than 1D or 2D generation methods across all metrics. However, this does not diminish the importance of 3D methods. On the contrary, 3D modelling is indispensable in SBDD, where understanding spatial relationships is crucial for evaluating and optimizing binding affinity to target proteins. Given these differences, we believe a direct comparison between DecompDPO and REINVENT 4 is not fair or meaningful.
>
> To better contextualize our work, we will include discussions in the revised version to clarify the unique contributions of 3D methods. While we will make efforts to explore comparative analyses, we hope to emphasize the inherent differences between 3D and 1D methods should be considered in the comparison. This distinction is critical for understanding the complementary roles of these approaches in addressing the unique challenges of SBDD.
>
> **A6:** Thank you for your efforts in reviewing our paper and your detailed suggestions! In our newly uploaded version, we have adjusted all these suggestions. Please let us know if you have any further questions.
>
>
> ### Remaining Concerns
> Thank you for your suggestion. We will update all these results in the revised version of paper.
>
> ### Supplementary Questions
> **Q1: $\beta$ schedule: shouldn't regularization be stronger at the end of the trajectory when molecular features are only refined but the global structure is more or less decided already? The optimization objective probably doesn't depend on small geometric details.**
>
> A1:
> In Direct Preference Optimization (DPO), we focus on optimizing the model parameters, not the molecules directly. This makes our model a generative one, different from traditional molecule optimization methods. Our goal is to adjust the distribution generated by DecompDiff to favor molecules with better properties.
>
> Diffusion-DPO uses the forward process to approximate sample molecules efficiently. During fine-tuning, there might be slight deviations, which could affect this approximation. The steps at the start of the trajectory affect those at the end, but not the other way around.
>
> Thus, we propose a beta schedule that emphasizes optimization at the end of the trajectory. This keeps the Diffusion-DPO approximation reliable while allowing enough flexibility to optimize the model effectively.
>
> **Q2: The DecompDPO model trained for 2 epochs seems to perform better. Why is it not trained even longer?**
>
> A2: Please refer to our response to Q3 in Weakness.
>
> Thank you once again for your detailed feedback. Please let us know if you have any further questions and suggestions.
>
> [1] Reidenbach, D. (2024). EvoSBDD: Latent Evolution for Accurate and Efficient Structure-Based Drug Design. In ICLR 2024 Workshop on Machine Learning for Genomics Explorations.

---

> ### Author Response · Authors · 2024-12-02
> **Gentle Reminder**
>
> Dear Reviewer HgKr,
>
> We are grateful for your insightful comments and the time you've taken to review our submission.
>
> As the deadline for the discussion phase approaches, we kindly ask if you could review our responses to your comments. Please let us know if there are any additional questions or concerns we can help clarify.
>
> We greatly appreciate your time and are committed to addressing any remaining concerns you may have.
>
> Sincerely,
>
> The Authors

---

> > ### Comment · Reviewer_HgKr · 2024-12-02
> > **Thank you for the feedback**
> >
> > Thanks again for the detailed responses. I trust the authors to include the new data in the updated version of the paper and will take this into account for my final evaluation.
> >
> > DecompDPO (2 epochs) performing better than KGDiff and the other baselines is indeed a good sign and I would recommend to focus more on this empirical performance gain than concerns about computational efficiency. Training 2 or even more epochs seems completely acceptable to me if it further improves the performance.
> >
> > I still do not see the claim of effectiveness of the multi-objective optimisation substantiated as long as spurious improvements based on correlations between the metrics lead to better results than direct optimisation.
> > As a final suggestion to address the above-mentioned discrepancy between the correlations on a local and global level, the authors could consider optimising for "docking efficiency" (Vina score divided by the number of heavy atoms) to reduce Vina's dependency on molecule size. Perhaps this helps to improve the effectiveness of the multi-objective training. This is probably less relevant for the current review round but could be tested in future work.

---

### Official Review · Reviewer_bSNV · 2024-10-30

**Soundness:** 3
**Presentation:** 3
**Contribution:** 2
**Rating:** 5
**Confidence:** 4

**Summary:**

In this paper, the authors propose DecompDPO, a method that applies Direct Preference Optimization (DPO) in a diffusion model for structure-based drug design (SBDD). In addition to the global DPO loss, which focuses on binding affinity as evaluated by AutoDock Vina, the paper introduces local DPO and a bond angle penalty to complement global DPO, aiming to improve ligand-protein binding affinity. DecompDPO achieves competitive performance on vina results compared to other selected baselines.

**Strengths:**

Thanks for the good work on exploring the DPO method in decomposed diffusion for SBDD. I found the work to have the following strengths:
1) The method achieved decent performance on vina score, vina min, and vina dock without significantly compromising on QED and SA scores compared with the baselines selected in the paper.
2) The paper is clearly written, easy to follow, and has well-designed figures.
3) The paper introduces the use of Local DPO for optimization, which goes beyond global DPO.

**Weaknesses:**

1. Performance claims in the paper

DecompDPO did outperform all selected baselines in terms of vina score, vina min, and vina dock, as mentioned in the paper. However, the paper did not benchmark against some other existing works that demonstrate better performance in improving the conditional sampling of diffusion models for SBDD. For example, KGDiff[1], a guidance-based method, has better performance on vina score, vina min, and vina dock. Both KGDiff and DecompDPO were benchmarked on the same test set with 100 test pockets and 100 samples per pocket. While I understand that vina results are not a precise measure of binding affinity and that not being SOTA in vina results does not necessarily mean the method is not good, I would still be cautious in stating that "DECOMPDPO achieves the highest score in Vina Minimize, Vina Dock, High Affinity, and Success Rate among all generative methods" in the paper.

2. Performance on Vina Score

DecompDPO shows good performance on vina min and vina dock, but the marginal improvement on vina score is small. Vina score directly reflects the quality of the sampled pose, while vina dock evaluates the quality of the pose sampled by the model and then optimized by AutoDock Vina. The results in the paper indicate that the improvement in the pose directly sampled from the generative model is minimal.

3. Effectiveness of Local DPO

Although local DPO is a novel idea compared to global DPO (e.g., AliDiff), Table 4 shows that the improvement brought by local DPO is marginal compared to global DPO. This raises questions about the effectiveness and real contribution of local DPO toward the overall improvement of local+global DPO.

Reference:
[1] KGDiff: towards explainable target-aware molecule generation with knowledge guidance

**Questions:**

Could the authors please provide the complete rates for the DecompDPO and DeompDiff baseline molecules? When running AutoDock Vina, in 100 samples for a pocket, only the successfully reconstructed molecules (usually fewer than 100 and sometimes around 50–60/100 for larger pockets) proceed to the vina docking process. This means the vina results in the table for all methods are actually calculated from fewer than 10,000 samples. It would be helpful to benchmark the method using this complete rate alongside the vina results. Thanks!

---

> ### Author Response · Authors · 2024-11-26
> **Answer to Weakness Part1**
>
> Thank you for your valuable feedback. Please see below for our responses to the comments.
>
> **Q1: Comparsion with KGDiff**
>
> A1: Thank you for your suggestion! We noticed that KGDiff reported the performance of binding knowledge guided generation benchmarked on CrossDocked2020. So we add the comparision with KGDiff in Single-Objective Optimization section.
>
> |            | Vina Score |           | Vina Min  |           | Vina Dock  |            | High Affinity |           | QED      |          | SA       |          | Diversity |          | Success Rate |
> | ---------- | ---------- | --------- | --------- | --------- | ---------- | ---------- | ------------- | --------- | -------- | -------- | -------- | -------- | --------- | -------- | ------------ |
> |            | mean       | median    | mean      | median    | mean       | median     | mean          | median    | mean     | median   | mean     | median   | mean      | median   | mean         |
> | IPDiff     | \-6.42     | \-7.01    | \-7.45    | \-7.48    | \-8.57     | \-8.51     | 69.5%         | 75.5%     | 0.52     | 0.53     | 0.60     | 0.59     | 0.74      | 0.73     | 17.7%        |
> | DecompDiff | \-5.96     | \-7.05    | \-7.60    | \-7.88    | \-8.88     | \-8.88     | 72.3%         | 87.0%     | 0.45     | 0.43     | 0.60     | 0.60     | 0.60      | 0.60     | 28.0%        |
> | KGDiff     | **−8.04**  | **−8.61** | **−8.78** | **−8.85** | −9.43      | −9.43      | 79.2%         | 87.0%     | **0.51** | **0.51** | 0.54     | 0.54     | \-        | \-       | \-           |
> | AliDiff    | \-7.07     | \-7.95    | \-8.09    | \-8.17    | \-8.90     | \-8.81     | 73.4%         | 81.4%     | 0.50     | 0.50     | 0.57     | 0.56     | **0.73**  | **0.71** | \-           |
> | DecompDPO  | \-6.31     | \-7.70    | \-8.49    | \-8.72    | **\-9.93** | **\-9.77** | **85.9%**     | **97.8%** | 0.48     | 0.46     | **0.66** | **0.66** | 0.65      | 0.65     | 43.0%        |
>
> As shown in the table, DecompDPO achieves the highest Vina Dock and High Affinity compared to AliDiff and KGDiff without sacrificing QED and SA, indicating its effectiveness. We will be more cautious about the pharsing when stating the performance of DecompDPO in both the Main Result section and the Single-Objective Optimization section.
>
> **Q2: Performance on Vina Score**
>
> A2: Thank you for bringing up this concern. We appreciate the opportunity to clarify our experimental choices and results regarding the Vina Score. In our experiment, Vina Minimize is selected as optimization objective rather than Vina Score, so the optimization does not prioritise Vina Score. There are several reasons for this choice: First, in SBDD, which is still in the early stages of pharmaceutical development, docking related metrics optimized by force field are crucial indicators for assessing the success of designed molecules, as it evaluates pose validity for downstream applications. Vina Minimize was chosen as optimization objective because it effectively reflects the binding affinity between the generated molecule and the protein without incurring high computational costs. Besides, in multi-objective optimization tasks, gradients from different objectives can conflict, potentially hindering overall performance. By focusing on Vina Minimize, which enables local optimization of atom positions, DecompDPO allows for greater flexibility and mitigates gradient conflicts.
>
> We want to emphasize that even though DecompDPO optimized towards Vina Minimize, which not prioritise the optimization towards Vina Score, the median Vina Score still showed a significant improvement compared to DecompDiff.
>
> To further illustrate the effectiveness of DecompDPO in optimizing binding affinity, including Vina Score, we extended the training for an additional epoch, totaling 30,000 fine-tuning steps, aligning with AliDiff. The experiment result is provided in Q3. As the result shows, DecompDPO with longer training achieves higher Vina Score compared to AliDiff, but with a slight trade-off in QED. These results suggest that DecompDPO can effectively optimize for higher binding affinity, but careful consideration is needed to balance trade-offs between optimization objective and general molecular properties during optimization.

---

> ### Author Response · Authors · 2024-11-26
> **Answer to Weakness Part2**
>
> **Q3: Effectiveness of Local DPO**
>
> A3: Thank you for pointing out this concern. Both Reviewer fiLe and Reviewer bSNV raised questions about the effectiveness of LocalDPO, and we appreciate the opportunity to clarify its advantages. In multi-objective optimization, gradients from different objectives may conflict, potentially hindering the optimization process and limiting the model's ability to improve all desired properties simultaneously. This conflict can make the benefits of LocalDPO less apparent in certain evaluations.
>
> To better illustrate the effectiveness of LocalDPO, we considered a more focused setting in binding affinity optimization. In this context, AliDiff can be regarded as employing a variant of GlobalDPO enhanced with Exact Energy Preference Optimization. To ensure a fair comparison, we extended the training of our model by an additional epoch, totaling 30,000 fine-tuning steps. The results are as follows:
> |                     | Vina Score |            | Vina Min   |            | Vina Dock   |             | High Affinity |            | QED      |          | SA       |          | Diversity |          | Success Rate |
> | ------------------- | ---------- | ---------- | ---------- | ---------- | ----------- | ----------- | ------------- | ---------- | -------- | -------- | -------- | -------- | --------- | -------- | ------------ |
> |                     | mean       | median     | mean       | median     | mean        | median      | mean          | median     | mean     | median   | mean     | median   | mean      | median   | mean         |
> | AliDiff             | \-7.07     | \-7.95     | \-8.09     | \-8.17     | \-8.90      | \-8.81      | 73.4%         | 81.4%      | 0.50     | 0.50     | 0.57     | 0.56     | **0.73**  | **0.71** | \-           |
> | DecompDPO - 1 epoch | \-6.31     | \-7.70     | \-8.49     | \-8.72     | \-9.93      | \-9.77      | 85.9%         | 97.8%      | **0.48** | **0.46** | 0.66     | 0.66     | 0.65      | 0.65     | 43.0%        |
> | DecompDPO - 2 epoch | **\-7.15** | **\-8.04** | **\-9.18** | **\-9.12** | **\-10.26** | **\-10.66** | **91.6%**     | **100.0%** | 0.47     | 0.44     | **0.67** | **0.67** | 0.69      | 0.69     | 45.4%        |
>
> As the results show, with LocalDPO optimizing Vina Minimize, our model achieves very significant improvements across binding affinity-related metrics, including Vina Score. The Complete Rate after 2 epochs of optimization is 77%. These results demonstrate LocalDPO's superior performance in binding optimization without sacrificing general molecular properties, highlighting its unique advantage in effectively navigating the optimization with decomposed perference.

---

> ### Author Response · Authors · 2024-11-26
> **Answer to Questions**
>
> **Q1: Provide complete rate for DecompDPO and DecompDiff**
>
> A1: Thank you for your question.  We report the Complete Rate, along with Novelty, Similarity, and Uniqueness in Table 7 in the original paper. We have further checked the sanity of generated molecules using PoseBuster. We computed the percentage of valid molecules (PB-Valid) checked by the PoseBuster [1] and reported the result in the following table:
>
> |            | Complete Rate | PB-Valid |
> | ---------- | ------------- | -------- |
> | DecompDiff | 72.82%        | 54.60%   |
> | DecompDPO  | 73.26%        | 56.56%   |
>
> As the table shows, DecompDPO achieves slightly improvement compared to DecompDiff in terms of Complete Rate and PB-Valid, confirming its effectiveness in generating valid molecules.
>
> [1] Buttenschoen, M., Morris, G. M., & Deane, C. M. (2024). PoseBusters: AI-based docking methods fail to generate physically valid poses or generalise to novel sequences. Chemical Science, 15(9), 3130-3139.

---

> ### Author Response · Authors · 2024-12-02
> **Gentle Reminder**
>
> Dear Reviewer bSNV,
>
> Thank you for the time and effort you've invested in evaluating our submission.
>
> In response to your valuable suggestions, we have added a comparative analysis with KGDiff and conducted additional experiments to demonstrate the effectiveness of DecompDPO in optimizing binding affinity.
>
> With the discussion phase coming to an end, we wanted to check if there are any further concerns we can address. If our responses have addressed your concerns, we kindly hope you might reconsider your evaluation of our submission.
>
> Sincerely,
>
> The Authors

---

### Official Review · Reviewer_fiLe · 2024-11-03

**Soundness:** 1
**Presentation:** 2
**Contribution:** 2
**Rating:** 3
**Confidence:** 4

**Summary:**

The authors propose to apply DPO to structure-based drug design. Authors formulate the DPO objective for diffusion models and suggest to apply it separately to different molecular fragments. Besides, authors propose to use the linear scheduling of the DPO regularization. Finally, authors incorporate physically-constrained optimisation to improve the geometry of the generated molecules. Overall, in my opinion, the work has interesting ideas and suggestions. However, the chosen evaluation methodology and the current results do not prove that the proposed techniques are beneficial at all. I discuss the weaknesses in details below, and believe there is a substantial scope for improvement. I will be happy to reconsider my decision after the rebuttal.

**Strengths:**

I find the proposed idea on the local DPO for molecular fragments very interesting and original and can potentially help in specific tasks. I also like the idea about constrained optimisation and believe it can be used in diffusion models in general to imporve the quality of the molecules. Overall, applying DPO to improve specific properties of the generated molecule makes total sense and can help generate high-quality molecules in cases when the quality of the training data is not sufficiently high.

**Weaknesses:**

**Major issues:**
1. LocalDPO looks like the main contribution of this paper. However, the evaluation of this component does not prove that the proposed mechanism is crucial for sampling molecules with optimised properties. In fact, based on Table 4, it looks like the model without LocalDPO solves the task equally well (the differences in the scores are too small). I can imagine that the chosen evaluation method does not highlight the advantages of the LocalDPO and its unique feature – locality. So my main question for which I struggle to find an answer in the paper: why would I use LocalDPO instead of the global one?

2. The same applies to the physical constrained optimisation. Tables 5 and 6 suggest that DPO doesn't improve the geometry of the molecules.

3. I am also confused about utilisation of different objectives in LocalDPO and GlobalDPO. In the implementation details, authors mention that they use QED, SA, and Vina Minimize Score. Does it apply to both local and global DPO components? If yes, then I do not understand the point of LocalDPO since neither QED nor SA are additive by fragments. If not, then all the main results concern only Vina optimisation, and I would recommend to make it more clear.

4. I am not entirely convinced by the claim regarding the improvement of molecular conformers in terms of energy (lines 428 - 458). First, in my opinion, the difference between distributions of values of DecompDiff and DecompDPO in Figures 4 (right), 5 and 6 is too small to say that DecompDPO is better. Second, why would it be better? The only geometric properties the model is optimized for are bond distances and bond angles. But I believe that having "correct" (i.e. within the distribution of the real data) bond distances and angles is only the necessary condition for the low conformational energy, but not sufficient. Besides, it is clear from Tables 5 and 6 that DecompDPO doesn't outperform DecompDiff in this regard. Therefore, the argument for improved molecular conformation feels unsubstantiated.

5. Single objective optimisation section (and Table 3) confuses me a lot:
* 5.1 Why do authors discuss improvements of QED and SA of the model that was optimized for affinity score only? How can these improvements be explained?
* 5.2 Why not to optimise separately for SA and for QED and then show how these scores improve after the corresponding optimisation? I believe that it is important to demonstrate the consistent ability to optimise different unrelated scores in separate experiments.
* 5.3 How does connectivity and validity depend on the fact that the model is optimised for Vina scores? To me this effect looks arbitrary and doesn't have any rationale behind (similar to what I discussed in 4.1). I agree that it is a good idea to discuss validity and connectivity, but I would suggest to do it in a different context – as a sanity check that demonstrates that the molecules generated by DecompDPO are in general adequate (for example, as an Appendix table).

**Minor comments and suggestions:**
1. Figure 4 (left) is not very informative – in my opinion it can be put to the appendix.
2. For clarity of the made arguments and to improve the overall structure of the paper, I believe that it would be beneficial to have another table in the main text with the key geometric scores: i.e. JS divergence between distances (values from the Figure 4 (left) legend), JS between angles, avg. conformer energies. At least, in my opinion, some comparison of angles should be in the main text, since authors provide equation (12).
3. Maybe we cannot see the DPO effects on bond distances and angles because $3\sigma$ (in equations 11 and 12) is a very loose threshold? What if to try just $\sigma$?
4. Line 472: Table 4 -> Table 3
5. Docking evaluation:
* 5.1. I believe that reporting scores after minimization or docking may not fully align with the goal of the proposed model, which is to generate 3D molecular structures that inherently adopt good binding conformations. Rather than presenting redocked and minimized structures, I would suggest evaluating the number of steric clashes between proteins and ligands, as this could provide a more direct assessment of the generated conformations’ suitability for binding.
 * 5.2. Due to the high correlation between the docking scores and molecule sizes, I would recommend reporting binding efficiency scores (i.e. docking score divided by the number of heavy atoms). I believe it provides a better signal of the potency of the generated molecules.
6. I would be curious to look at the validity and connectivity metrics - as an overall quality assessment of the DecompDPO molecules. Also would interesting to see the percentage of the samples that pass PoseBusters filters.

**Questions:**

1. Figure 4 (right): what does fragment size on x-axis mean? (Average?) number of atoms in the fragments? Or (average?) number of fragments in the molecules?
2. How are the fragments without rotatable bonds computed (in the section about MMFF evaluation)?
3. It would be interesting to look at the distributions of the scores in winning and loosing examples in the created dataset for preference alignment.
4. Do I understand correctly that linear $\beta$-schedule means that in cases when t is large enough (i.e., closer to the beginning of the denoising trajectory) the DPO loss is scaled by a large number? And does it mean that effectively the model learns to align only the later stages of the denoising process?
5. Did authors try to use (11) and (12) as penalties while training the diffusion model itself? Looks like it is a nice additional penalty in general
6. How do authors select the size of the sampled molecules? Is the distribution of sizes in samples the same with and without DPO?

---

> ### Author Response · Authors · 2024-11-26
> **Answer to Weaknesses Part1**
>
> Thank you for your constructive feedback. Please see below for our responses to the comments.
>
> **Q1: In Table 4, the contribution of LocalDPO is not evident. The current evaluation does not highlight the advantages of LocalDPO and its unique feature – locality. Why should LocalDPO be used instead of GlobalDPO?**
>
> A1: Thank you for bringing up this concern. We appreciate the opportunity to further clarify the advantages of LocalDPO. In multi-objective optimization, the gradients for different objectives may conflict with each other, potentially hinder the optimization process and limit the model's ability to improve all desired properties effectively, resulting in the advantage of LocalDPO not highlighted very significantly. The effectiveness of LocalDPO could be better illustrated by comparison with AliDiff, which can be regarded as employing a variant of GlobalDPO enhanced with Exact Energy Preference Optimization to improve results. To ensure a fair comparison, we extended the training of our model by an additional epoch, totaling 30,000 fine-tuning steps. The results are shown as follows:
> |                     | Vina Score |        | Vina Min |        | Vina Dock |         | High Affinity |        | QED  |        | SA   |        | Diversity |        | Success Rate |
> | ------------------- | ---------- | ------ | -------- | ------ | --------- | ------- | ------------- | ------ | ---- | ------ | ---- | ------ | --------- | ------ | ------------ |
> |                     | mean       | median | mean     | median | mean      | median  | mean          | median | mean | median | mean | median | mean      | median | mean         |
> | AliDiff             | \-7.07     | \-7.95 | \-8.09   | \-8.17 | \-8.90    | \-8.81  | 73.4%         | 81.4%  | 0.50 | 0.50   | 0.57 | 0.56   | 0.73      | 0.71   | \-           |
> | DecompDPO - 1 epoch | \-6.31     | \-7.70 | \-8.49   | \-8.72 | \-9.93    | \-9.77  | 85.9%         | 97.8%  | 0.48 | 0.46   | 0.66 | 0.66   | 0.65      | 0.65   | 43.0%        |
> | DecompDPO - 2 epoch | \-7.15     | \-8.04 | \-9.18   | \-9.12 | \-10.26   | \-10.66 | 91.6%         | 100.0% | 0.47 | 0.44   | 0.67 | 0.67   | 0.69      | 0.69   | 45.4%        |
>
> As the result shows, with LocalDPO optimizing Vina Minimize, our model achieves significant improvements across binding affinity related metrics while not sacrifycing molecular properties. This demonstrates LocalDPO's superior performance in binding optimization and highlights its unique advantage in effectively navigating the optimization with decomposed perference.
>
> **Q2: Tables 5 and 6 suggest that DPO doesn't improve the geometry of the molecules.**
>
> A2: We appreciate the opportunity to further clarify the motivation of our physically-constrained optimization. Indeed, as we stated in Section 3.3, the purpose of introducing physics-informed energy terms in preference alignment is to maintain reasonable molecular conformations instead of optimizing the geometry of generated molecules. Out of this consideration, we only penalize unreasonable conformations that digress away from $3\sigma$ threshold. In Table 5 and 6, we demonstrate that the physically-constrained optimization generally achieves comparable performance with DecompDiff in terms of molecular conformation, which is align with the motivation.
>
> **Q3: Calrify the objectives used for QED, SA, and Vina Minimize Score regarding LocalDPO and GlobalDPO**
>
> A3: As stated in Section 3.2 DECOMPDPO, we use LocalDPO or GlobalDPO according to each optimization objective's decomposability in multi-objective optimization. The total loss is a weighted sum over all loss terms for different objectives. In experiment, we used GlobalDPO for QED and SA since they are non-decomposable, and used LocalDPO for Vina Minimize optimization.
>
> **Q4: The argument for improved molecular conformation is unsubstantiated.**
>
> A4: Thank you for your detailed review and highlighting this point. As we stated in Q2, as preference alignment inevitably optimizes model distribution towards high rewards and away from the original distribution, our initial purpose of introducing physical constraints is to maintain reasonable conformations during optimization. For the sentence in line 431, our intent is to illustrate the experiment result we observed from Figure 4 (right), with DecompDPO achieves lower energy difference for rigid fragments with more heavy atoms. We will be more cautious about our phrasing and have revised this sentence in the paper.

---

> ### Author Response · Authors · 2024-11-26
> **Answer to Weaknesses Part2**
>
> **Q5.1: How to explain the improvements of QED and SA when optimizing for binding affinity?**
>
> A5.1: Thank you for your insightful questions. Typically, optimizing a model toward a specific objective like binding affinity can lead to trade-offs with other molecular properties such as QED and SA. For instance, when AliDiff is optimized for binding affinity, both QED and SA scores tend to decrease. However, we observed that when DecompDPO is optimized solely for binding affinity, the QED and SA scores actually improve, which is unusual in single-objective optimization. To explain this, we calculated the Pearson correlation coefficients between the differences in Vina Minimize scores and the differences in QED and SA scores for each pair in our dataset. We found that the correlations between differences in the decomposed arms' Vina Minimize and the corresponding molecules' differences in QED and SA are -0.12 and -0.11, respectively. These statistics suggest that the improvements in QED and SA are potentially due to the benefits of LocalDPO, where the flexibility introduced by the decomposed chemical space alleviates conflicts between SA and binding affinity, allowing the model to navigate the chemical space effectively even when optimizing for binding affinity. Besides, in multi-objective optimization, the gradients of different objectives may conflict with each other, leading to less effective results, which could also affect the optimization performance of QED and SA.
>
> **Q5.2: The single objective optimization result for QED and SA.**
>
> A5.2: Thanks for your questions. We have further conducted experiments optimizing QED and SA separately, the results are shown below. As QED and SA are non-decomposable, we used GlobalDPO for the optimizaiton.
> |                      | Vina Score |            | Vina Min   |            | Vina Dock  |            | High Affinity |           | QED      |          | SA       |          | Diversity |        | Success Rate |
> | -------------------- | ---------- | ---------- | ---------- | ---------- | ---------- | ---------- | ------------- | --------- | -------- | -------- | -------- | -------- | --------- | ------ | ------------ |
> |                      | mean       | median     | mean       | median     | mean       | median     | mean          | median    | mean     | median   | mean     | median   | mean      | median | mean         |
> | DecompDiff           | \-5.96     | \-7.05     | \-7.60     | \-7.88     | \-8.88     | \-8.88     | 72.3%         | 87.0%     | 0.45     | 0.43     | 0.60     | 0.60     | 0.60      | 0.60   | 28.0%        |
> | DecompDPO - QED      | \-6.06     | \-7.27     | \-8.05     | \-8.27     | \-8.99     | \-9.36     | 81.5%         | 92.6%     | **0.48** | **0.47** | 0.65     | 0.65     | 0.63      | 0.63   | 37.8%        |
> | DecompDPO - SA       | \-6.16     | \-6.90     | \-7.88     | \-8.02     | \-8.90     | \-8.85     | 72.8%         | 87.2%     | **0.48** | 0.46     | **0.66** | **0.66** | 0.61      | 0.60   | 34.0%        |
> | DecompDPO - Vina Min | **\-6.31** | **\-7.70** | **\-8.49** | **\-8.72** | **\-9.93** | **\-9.77** | **85.9%**     | **97.8%** | **0.48** | 0.46     | **0.66** | **0.66** | 0.65      | 0.65   | **43.0%**    |
>
> As shown in the experimental result, DecompDPO achieves 6.7% and 10% in the corresponding objective for QED and SA optimization, respectively. As we discussed in Q5.1, there are positive side effects between QED, SA, and Vina arise from the correlation in our dataset, so the binding affinity is slightly improved. Overall, the single objective optimization experiments demonstrate the effectiveness of DecompDPO, especially in the setting that using LocalDPO to optimize Vina Minimize.
>
> **Q5.3: Discussion of connectivity and validity in Single-Objective Optimization.**
>
> A5.3: Thank you for your question. In Section 4.3 Single-Objective Optimization, we reported the Complate Rate to demonstrate that the improvement of molecular properties is not at the expense of the Complete Rate. In Table 7, we have reported the Complate Rate, Novelty, Similarity, and Uniqueness of DecompDPO to demonstrate its ability in designing novel and valid molecules.
>
> **Q6: Put Figure 4 (left) to appendix. Typo in Line 472: Table 4 -> Table 3.**
>
> A6: Thank you for pointing this out! We will carefully proofread our manuscript and revise these.

---

> ### Author Response · Authors · 2024-11-26
> **Answer to Weaknesses Part3**
>
> **Q7: Have another table in the main text with the key geometric scores: JS divergence between distance, JS divergence between angles, Avg. conformer energies.**
>
> A7: Thank you for your suggestion. We have further summarized JSD of pairwise distance distributions provided in Figure 4 (left), and RMSD and energy difference of rigid fragments and the whole molecule provided in Figure 4 (right), 6, 7, 8 in the following table:
> |            | JSD Dist | Energy Diff - rigid fragments | RMSD - rigid fragments | Energy Diff - Mol | RMSD - Mol |
> | ---------- | -------- | ----------------------------- | ---------------------- | ----------------- | ---------- |
> | Pocket2Mol | 0.14     | **31.18**                     | 0.12                   | **185.14**        | **0.75**   |
> | TargetDiff | 0.09     | 1355.94                       | 0.13                   | 6116.37           | 1.02       |
> | IPDiff     | 0.08     | 1459.45                       | 0.14                   | 21431.71          | 1.04       |
> | DecompDiff | **0.07** | 39.39                         | 0.13                   | 8833.80           | 1.10       |
> | DecompDPO  | **0.07** | 42.49                         | **0.11**               | 976.33            | 1.11       |
>
> We have added Table 7 in the Appendix to provide numerical evidence in addition to the distributional evidence for molecular conformations and will adjust the layout in the future version. For JSD between bond distance and angles, we provide the evaluation results in Table 5 and 6 in Appendix.
>
> **Q8: Try physical constraints with $\sigma$ threshold**
>
> A8: Thank you for your insightful question. We have further conducted an experiment using physical constraints with $\sigma$ threshold for multi-objective optimization.
>
> |                     | Vina Score |        | Vina Min |        | Vina Dock |        | High Affinity |        | QED  |        | SA   |        | Diversity |        | Success Rate |
> | ------------------- | ---------- | ------ | -------- | ------ | --------- | ------ | ------------- | ------ | ---- | ------ | ---- | ------ | --------- | ------ | ------------ |
> |                     | mean       | median | mean     | median | mean      | median | mean          | median | mean | median | mean | median | mean      | median | mean         |
> | DecompDPO - sigma   | \-6.05     | \-7.21 | \-7.89   | \-8.16 | \-9.08    | \-9.22 | 78.0%         | 96.0%  | 0.48 | 0.46   | 0.64 | 0.64   | 0.61      | 0.61   | 35.0%        |
> | DecompDPO - 3 sigma | \-6.10     | \-7.22 | \-7.93   | \-8.16 | \-9.26    | \-9.23 | 78.2%         | 95.2%  | 0.48 | 0.45   | 0.64 | 0.64   | 0.62      | 0.62   | 36.2%        |
>
> |                     | JSD Dist | Energy Diff - rigid fragments | RMSD - rigid fragments | Energy Diff - Mol | RMSD - Mol |
> | ------------------- | -------- | ----------------------------- | ---------------------- | ----------------- | ---------- |
> | DecompDPO - sigma   | 0.07     | 31.00                         | 0.11                   | 857.12            | 1.12       |
> | DecompDPO - 3 sigma | 0.07     | 42.49                         | 0.11                   | 976.33            | 1.11       |
>
> As we emphasized before, the physical constraints are proposed for penalizing unreasonable conformations during preference alignment. The results show that the stricter physical constraints with $\sigma$ threshold basically does not affect optimization performance, with a very slight drop in binding related metrics, and leads to lower energy difference for both rigid fragments and generated molecules compared to $3\sigma$ threshold.

---

> ### Author Response · Authors · 2024-11-26
> **Answer to Weaknesses Part4**
>
> **Q9: Report evaluation of the number of steric clashes. Report binding efficiency scores such as docking scores divided by the number of heavy atoms.**
>
> A9: Thank you for your suggestion. To provide a more comprehensive evaluation, we benchmarked DecompDPO using PoseCheck [1]. We reported the quartiles of Strain Energy (SE), the average number of Steric Clashes, and the number of key interactions, i.e., Hydrogen Bond Donors (HB Donors), Hydrogen Bond Acceptors (HB Acceptors), van-der Waals contacts (vdWs) and Hydrophobic interactions (Hydrophobic).
> |            | SE     |        |        | Clash | HB Donors | HB Acceptor | Hydrophobic | vdWs  |
> | ---------- | ------ | ------ | ------ | ----- | --------- | ----------- | ----------- | ----- |
> |            | 25%    | 50%    | 75%    | Avg.  | Avg.      | Avg.        | Avg.        | Avg.  |
> | Reference  | 34     | 107    | 196    | 5.51  | 0.87      | 1.42        | 5.06        | 6.61  |
> | TargetDiff | 369    | 1243   | 13871  | 10.84 | 0.63      | 0.98        | 5.43        | 7.92  |
> | DecompDiff | 161.87 | 354.28 | 802.43 | 15.42 | 0.59      | 1.48        | 7.96        | 11.06 |
> | DecompDPO  | 141.14 | 322.82 | 723.90 | 15.05 | 0.43      | 1.31        | 8.25        | 11.01 |
>
> As shown in the table, DecompDPO achieves better Strain Energy compared to TargetDiff and DecompDiff. Regarding Steric Clashes, although DecompDPO performs slightly better than DecompDiff, we acknowledge that there are a few targets with extreme clashes due to anomalies beta priors detected by auto process. Addressing this issue is part of our future work.
>
> In terms of binding efficiency, while the number of heavy atoms in molecules relates to binding affinity, their correlation is not strictly linear. To better illustrate the potency of the generated molecules, we further drew scatter plots of heavy atom numbers versus Vina Dock scores for DecompDPO and baseline models. As Figure 10 in Appendix shows, DecompDPO generally achieves better Vina Dock scores compared to DecompDiff for molecules with the same number of heavy atoms.
>
> **Q10: Report validity, connectivity, and percentage of the samples that pass PoseBusters filters.**
>
> A10: Thank you for your question. In the original paper, we report the Complete Rate, which is defined as the percentage of valid and connected molecules, in Table 7. To further evaluate the performance of DecompDPO, we computed the percentage of valid molecules (PB-Valid) checked by the PoseBusters [2].
>
> |            | Complete Rate | PB-Valid |
> | ---------- | ------------- | -------- |
> | DecompDiff | 72.82%        | 54.60%   |
> | DecompDPO  | 73.26%        | 56.56%   |
>
> As the table shows, DecompDPO achieves slightly improvement compared to DecompDiff in terms of Complete Rate and PB-Valid.
>
> [1] Harris, C., Didi, K., Jamasb, A., Joshi, C., Mathis, S., Lio, P., & Blundell, T. (2023). Posecheck: Generative models for 3d structure-based drug design produce unrealistic poses. In NeurIPS 2023 Generative AI and Biology (GenBio) Workshop.
>
> [2] Buttenschoen, M., Morris, G. M., & Deane, C. M. (2024). PoseBusters: AI-based docking methods fail to generate physically valid poses or generalise to novel sequences. Chemical Science, 15(9), 3130-3139.

---

> ### Author Response · Authors · 2024-11-26
> **Answer to Questions Part1**
>
> **Q1 & Q2: How are the fragments without rotatable bonds computed? In Figure 4 (right), what does fragment size on x-axis mean?**
>
> A1: Thank you for your question. In evaluating the median RMSD and energy difference of rigid fragments, we first fragmentation each molecule by rotatable bonds. Then we group rigid fragments by their nubmer of heavy atoms and calculate the median RMSD and energy difference. In Figure 4 (right), the x-axis is the number of heavy atoms contained in rigid fragments.
>
> **Q3: Distributions of the scores in winning and losing examples in the created dataset for preference alignment.**
>
> A3: Thank you for your question. We have further provided winning and losing molecules' distribution of QED, SA, and Vina Minimize in Figure 13 in the Appendix.
>
> **Q4: Clarify the beta schedule.**
>
> A4: Thank you for your question. Yes, in our linear $\beta$-schedule, we set a large $\beta$ when $t$ is large to ensure strong regularization from the reference model during the initial stages, maintaining stability and adherence to the learned data distribution. As $t$ decreases, moving toward the later stages of denoising, we gradually reduce $\beta$. This smaller $\beta$ allows for more aggressive optimization guided by the specific objectives, enabling the model to align more closely with the desired properties. This strategy effectively balances the influence of the reference model with the optimization goals throughout the denoising process.

---

> ### Author Response · Authors · 2024-11-26
> **Answer to Questions Part2**
>
> **Q5: Use physics-informed energy terms as additional penalty loss**
>
> A5: Thank you for your question. We had conducted an experiment where we used physics-informed energy terms as additional penalty losses during the fine-tuning. In this setup, the physics-informed terms were directly applied as penalty losses to the output of the fine-tuning model, rather than being used as constraints for penalizing the DPO rewards.
> |                          | Vina Score |        | Vina Min |        | Vina Dock |        | High Affinity |        | QED  |        | SA   |        | Diversity |        | Success Rate |
> | ------------------------ | ---------- | ------ | -------- | ------ | --------- | ------ | ------------- | ------ | ---- | ------ | ---- | ------ | --------- | ------ | ------------ |
> |                          | mean       | median | mean     | median | mean      | median | mean          | median | mean | median | mean | median | mean      | median | mean         |
> | DecompDiff               | \-5.96     | \-7.05 | \-7.60   | \-7.88 | \-8.88    | \-8.88 | 72.3%         | 87.0%  | 0.45 | 0.43   | 0.60 | 0.60   | 0.60      | 0.60   | 28.0%        |
> | DecompDPO - penalty loss | \-6.27     | \-6.97 | \-7.84   | \-7.82 | \-9.15    | \-9.03 | 74.4%         | 93.1%  | 0.48 | 0.45   | 0.63 | 0.63   | 0.60      | 0.62   | 31.1%        |
> | DecompDPO                | \-6.10     | \-7.22 | \-7.93   | \-8.16 | \-9.26    | \-9.23 | 78.2%         | 95.2%  | 0.48 | 0.45   | 0.64 | 0.64   | 0.62      | 0.62   | 36.2%        |
>
> As the result shows, using physics-informed energy terms as penalty losses leads to improvements over the baseline model, but does not perform as well as our proposed method where these terms are used as reward penalties in optimization. We also compared the models in terms of molecular conformation:
> |                          | JSD Dist | Energy Diff - rigid fragments | RMSD - rigid fragments | Energy Diff - Mol | RMSD - Mol |
> | ------------------------ | -------- | ----------------------------- | ---------------------- | ----------------- | ---------- |
> | DecompDPO - penalty loss | 0.07     | 21.64                         | 0.11                   | 677.39            | 1.09       |
> | DecompDPO                | 0.07     | 42.49                         | 0.11                   | 976.33            | 1.11       |
>
> Using physics-informed energy terms as penalty losses resulted in lower energy differences and slightly better RMSD values, indicating better preservation of molecular conformations. We hypothesize that this occurs because applying the energy terms as penalty losses more directly penalizes unreasonable outputs, but at the expense of reduced optimization effectiveness toward the desired objectives.
>
> **Q6: How do authors select the size of the sampled molecules? Is the distribution of sizes in samples the same with and without DPO?**
>
> A6: Thank you for your question. For both molecule generation and molecular optimization, we employ the same Opt Prior used in DecompDiff, ensuring the same samples size with and without DPO. Opt Prior is defined as a mixture of Ref Prior, which is determined by the reference ligand, and Pocket Prior, which is defined by a prior generation algorithm using AlphaSpace2 [3] toolkit, depending on whether Ref Prior passes the Success threshold (QED > 0.25, SA > 0.59, Vina Dock < −8.18). For Ref Prior, the generated molecule size is the same as the reference molecule, and for Pocket Prior, the generated molecule size is determined by the size of the detected pocket. We have added the details of molecule size selection in the Experimental Details section in Appendix.
>
> [3] Katigbak, J., Li, H., Rooklin, D., & Zhang, Y. (2020). AlphaSpace 2.0: representing concave biomolecular surfaces using β-clusters. Journal of chemical information and modeling, 60(3), 1494-1508.

---

> ### Author Response · Authors · 2024-12-02
> **Gentle Reminder**
>
> Dear Reviewer fiLe,
>
> We sincerely appreciate your detailed and constructive comments on our submission.
>
> In our response, we have further clarified the motivation and advantages of DecompDPO, provided a more comprehensive evaluation, and conducted additional experiments as you suggested.
>
> As the discussion phase is nearing its conclusion, we would greatly appreciate any additional feedback you might have. If we have properly addressed your concerns, we kindly hope you might reconsider your evaluation of our submission.
>
> Sincerely,
>
> The Authors

---

### Meta-Review · Area_Chair_UHDa · 2024-12-22

**Metareview:**

In this paper, authors propose DecompDPO, a method that applies Direct Preference Optimization (DPO) to diffusion models for structure-based drug design (SBDD). The key technical contributions include: 1) Decomposed preference optimization that operates at both global molecular and local substructure levels, 2) Physics-informed energy terms to maintain reasonable molecular conformations, and 3) A framework that can be used for both fine-tuning pretrained models and post-generation optimization. The authors evaluate their method on the CrossDocked2020 benchmark and report improvements in binding affinity metrics while maintaining drug-likeness properties.

The work has several strengths as follows. The paper addresses an important problem in SBDD by developing preference-guided optimization of generated molecules. The application of DPO to molecular generation is novel and timely. The decomposed approach that operates at different granularities (global vs local) is technically interesting. The authors provide extensive ablation studies and experimental results comparing against relevant baselines. The physics-informed constraints represent a thoughtful addition to maintain molecular validity during optimization.

However, there are several weaknesses for the work as follows. The empirical improvements demonstrated by the method are notably modest, particularly for the core metrics that the approach explicitly aims to optimize. Reviewer HgKr emphasized this point, noting that the multi-objective optimization sometimes performs worse than single-objective optimization, which raises fundamental questions about the effectiveness of the proposed methodology. Reviewer koEE highlighted that the heavily emphasized Vina Dock metric may not be the most meaningful measure, as it involves re-docking that operates independently of the generated 3D structure. The raw Vina Score shows only a marginal improvement of 0.14 kcal/mol over the baseline DecompDiff. The analysis remains incomplete in several crucial aspects - multiple reviewers requested additional experiments and comparisons that were not fully addressed in the rebuttal, including comparisons with 2D methods like TacoGFN and a more thorough investigation of why DPO appears to degrade performance when applied to different architectures like TargetDiff. A significant practical limitation lies in the validity issues - the method achieves lower complete rates compared to some baselines, with up to 35% of generated molecules being invalid or disconnected. While the authors provided some explanation in their rebuttal, this remains a concerning limitation for practical applications. These weaknesses collectively suggest that while the underlying ideas may have merit, the current implementation and analysis fall short of demonstrating a clear advance in the field of structure-based drug design.

While the paper presents interesting ideas and makes a reasonable attempt at improving molecular generation through preference optimization, the empirical results do not convincingly demonstrate the effectiveness of the proposed method. The modest gains in primary metrics, coupled with concerning regressions in some scenarios, suggest the approach may need substantial refinement before it represents a clear advance in the field. The authors' rebuttal, while thorough, did not fully address core methodological concerns raised by reviewers regarding evaluation metrics and architectural choices. The incomplete rates and limited improvement over simpler baselines make it difficult to justify acceptance at this time.

For future submissions, I recommend: 1) More thorough investigation of why multi-objective optimization underperforms single-objective in some cases, 2) Addressing the validity rate issues, 3) Expanding comparisons to include both 2D and 3D baselines while clearly articulating the tradeoffs, and 4) Providing more rigorous analysis of the method's behavior when applied to different architectures.

**Additional Comments On Reviewer Discussion:**

See the comments above.

---

### Decision · Program_Chairs · 2025-01-22

Reject